# Robust Cross-Modal Retrieval via Generative Semantic Refinement and Exclusion-Guided Adaptation

**Qin Yang** [1]  **Xin Wei** [1]  **Yanjia Li** [2]  **Jiasun Feng** [3]  **Mingrui Zhu** [1]  **Nannan Wang** [1]  **Xinbo Gao** [1]

## Abstract

Vision-Language Pre-trained (VLP) models are vulnerable to real-world query noise. Current cross-modal Test-Time Adaptation (TTA) methods often rely on high-confidence predictions, which induces confirmation bias and neglects the informative signals in ambiguous Low-Confidence Queries. To address this, we propose Generative Semantic Refinement and Exclusion-Guided Adaptation (ReEx), a robust retrieval framework that extends adaptation to the entire query stream. Specifically, textual structural noise is rectified by a Generative Semantic Refinement (GSR) module, which employs Confidence-Guided Dynamic Fusion to anchor LLM-based repairs and prevent semantic drift. To exploit ambiguous data, adaptation is driven by Exclusion-Guided Proxy Contrastive Learning (EPCL), which imposes negative constraints via Exclusion Sets of unlikely candidates. Experimental results on COCO-C and Flickr-C demonstrate that ReEx consistently outperforms existing TTA methods, achieving significant robustness gains with a justifiable computational trade-off. Code will be made available at https://github.com/qinyxdu/ReEx.

## 1. Introduction

Cross-modal retrieval maps visual and textual inputs into a unified semantic space, enabling applications such as image-text search and recommendation systems. While Vision-Language Pre-trained (VLP) models (Radford et al., 2021; Jia et al., 2021; Li et al., 2022; 2023; Liu et al., 2023) demonstrate strong performance on standard benchmarks, they implicitly assume that test queries share the same distribution

---

[1]Xidian University [2]Xi'an Jiaotong University [3]Video Rebirth. Correspondence to: Xin Wei <weixin@xidian.edu.cn>, Nannan Wang <nnwang@xidian.edu.cn>.

*Proceedings of the 43$^{rd}$ International Conference on Machine Learning*, Seoul, South Korea. PMLR 306, 2026. Copyright 2026 by the author(s).

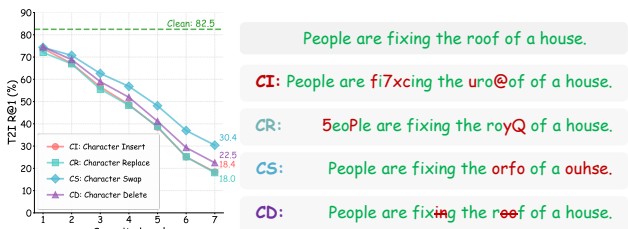

*(a)* Vulnerability of VLP models

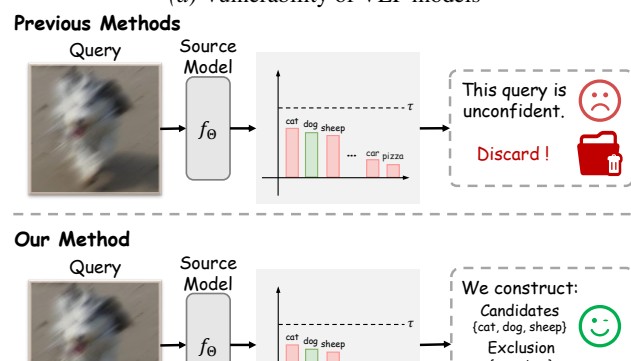

*(b)* Proposed strategy for Low-Confidence Queries

*Figure 1.* **Motivation and conceptual comparison. (a)** Impact of structural noise on retrieval. Even minor typing mistakes (CI, CR, CS, CD) alter tokenization, leading to a drastic drop in Recall@1. **(b)** Strategies for Low-Confidence Queries. Traditional methods view ambiguity as unreliable and discard such queries (red bin). In contrast, ReEx constructively utilizes them by defining an **Exclusion Set** (the "must not be" set, *e.g.*, excluding "car" and "pizza"), thereby narrowing the search space for the target query.

as the training data. In practice, however, this assumption often does not hold. As shown in Fig. 1(a), BLIP (Li et al., 2022) exhibits noticeable performance degradation when queries contain common perturbations such as character insertions or swaps. Unlike humans, who are robust to such typos, VLP models are hypersensitive to discrete token perturbations: even minor variations can alter tokenization and drastically shift embeddings, leading to retrieval failures.

Test-Time Adaptation (TTA) (Liang et al., 2025; Wang et al., 2021) offers a paradigm to update models during inference without accessing source data, making it ideal for handling

query shifts. Recent work TCR (Li et al., 2025) applies TTA to cross-modal retrieval by regulating query modality uniformity and the modality gap, while minimizing entropy on high-confidence queries. Despite its effectiveness, we observe that this strategy faces challenges, particularly regarding structural noise. First, for textual queries, standard TTA methods operate directly on raw, potentially corrupted embeddings. Since discrete perturbations (*e.g.*, typos) can disrupt tokenization and shift semantic representations, optimizing the model to align with these corrupted features may limit adaptation quality. Second, Low-Confidence Queries are typically discarded as unreliable. However, we posit that these ambiguous queries contain valuable negative certainty: even when the exact match is uncertain, the model can often reliably exclude irrelevant candidates (as illustrated in Fig. 1(b)). Consequently, current approaches tend to overlook structural rectification for noisy text and underutilize the informative exclusion constraints in ambiguous queries.

To address these challenges, we propose ReEx, a unified framework designed to leverage the full potential of the query stream. First, to establish a reliable foundation for adaptation, we introduce the **Generative Semantic Refinement (GSR)** module. Since adaptation on structurally corrupted features is often suboptimal, GSR serves as a pivotal rectification step for textual queries. It employs a **Confidence-Guided Dynamic Fusion** mechanism to anchor LLM-based repairs to original token-level cues, promoting semantic consistency before adaptation begins. With reliable features in place, the core of our framework lies in **Exclusion-Guided Proxy Contrastive Learning (EPCL)**. This module represents a distinct departure from standard approaches that rely solely on positive certainty. By defining an Exclusion Set of unlikely candidates based on predictive ambiguity, EPCL imposes informative negative constraints. This allows the model to refine decision boundaries by pushing queries away from irrelevant semantics, effectively mining valid supervision from the entire data distribution—turning the typically discarded Low-Confidence Queries into a rich source of adaptive signals.

Our contributions are summarized as follows:

**(1)** We identify two intrinsic challenges in existing cross-modal TTA: the reliance on structurally corrupted representations for textual adaptation and the underutilization of negative certainty in Low-Confidence Queries.
**(2)** We propose ReEx, a unified framework that integrates Generative Semantic Refinement (GSR) to rectify structural textual noise via confidence-guided fusion, and Exclusion-Guided Proxy Contrastive Learning (EPCL) to mine informative negative constraints from ambiguous queries.
**(3)** Extensive experiments on the standard robustness benchmarks, COCO-C and Flickr-C, demonstrate that ReEx establishes a new state-of-the-art. Notably, it surpasses the strong

baseline TCR by significant margins of **2.3%** (T2I) and **2.4%** (I2T) on Flickr-C. Ablation studies further validate the critical synergy between GSR's structural rectification and EPCL's negative mining.

## 2. Related Works

### 2.1. Domain Adaptation for Cross-Modal Retrieval

Cross-modal retrieval maps visual and textual inputs into a unified semantic space. While Vision-Language Pre-trained (VLP) models (Radford et al., 2021; Li et al., 2022; 2023) demonstrate strong zero-shot capabilities, they often struggle when test queries deviate from the training distribution. To mitigate such shifts, Unsupervised Domain Adaptation (UDA) methods (Munro et al., 2021; Hao et al., 2023; Han et al., 2024) utilize both labeled source data and unlabeled target data. Common strategies rely on pseudo-labeling to mine reliable alignments (Munro et al., 2021; Hao et al., 2023; Han et al., 2024; Chen et al., 2021), statistical alignment to reduce domain discrepancies (Peng & Chi, 2019), or prototype-based mutual information maximization (Liu et al., 2021a). However, a critical bottleneck of these methods is their reliance on source data access during adaptation. This dependency, combined with the high computational cost of offline training, limits their applicability in real-time scenarios like search engines. Consequently, practical applications often demand online adaptation on sequential query streams, where source data is unavailable due to privacy or storage constraints.

### 2.2. Test-Time Adaptation

Test-Time Adaptation (TTA) (Liang et al., 2025; Schneider et al., 2020; Sun et al., 2020; Wang et al., 2021; Liang et al., 2020) aims to mitigate distribution shifts by updating model parameters online during inference, eliminating the need for source data access. Early approaches (Sun et al., 2020; Liu et al., 2021b) relied on self-supervised auxiliary tasks introduced during training. To avoid modifying the training phase, subsequent works focused on efficient adaptation strictly during testing. For instance, TENT (Wang et al., 2021) minimizes prediction entropy by updating Batch Normalization (BN) statistics, while SHOT (Liang et al., 2020) optimizes feature extractors using pseudo-labels with a frozen classifier. Further advancements have addressed stability and efficiency: SAR (Niu et al., 2023) and CoTTA (Wang et al., 2022) introduce stochastic strategies to prevent model collapse, while EATA (Niu et al., 2022) and T3A (Iwasawa & Matsuo, 2021) employ sample filtering and prototype-based memory to reduce computational overhead.

While effective for classification, extending TTA to retrieval presents unique challenges. Unlike classification, which

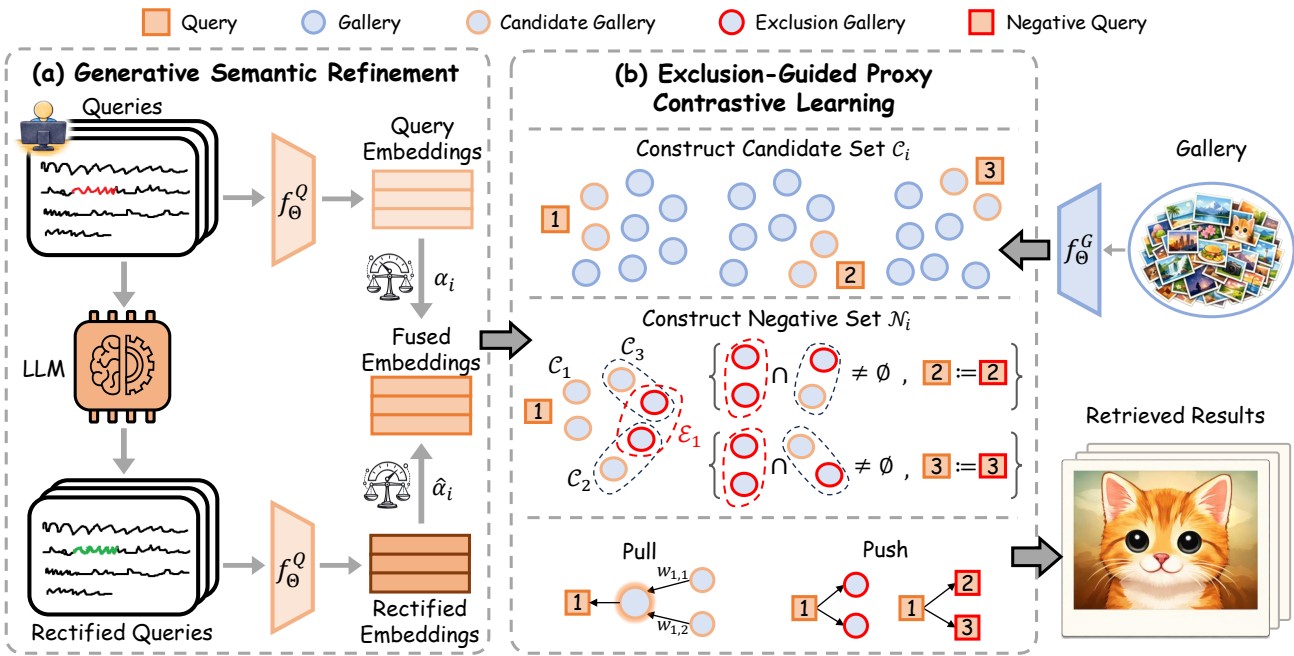

*Figure 2.* **Overall architecture of ReEx.** The framework is composed of two modules working in synergy. **(a) GSR Module:** It operates exclusively on text, fusing raw and LLM-rectified text embeddings to correct structural noise. The balance scales visualize the **Confidence-Guided Dynamic Fusion**, which calculates the interpolation weight $\alpha_i$ based on retrieval reliability. **(b) EPCL Module:** It categorizes gallery samples into the **Candidate Set** ($\mathcal{C}_i$, orange nodes) and the **Exclusion Set** ($\mathcal{E}_i$, red nodes). The adaptation objective (bottom right) leverages these sets to *pull* reliable candidates closer while decisively *pushing* away ambiguous exclusion samples.

optimizes discrete decision boundaries, retrieval requires preserving the relative similarity structure of the metric space. This complexity is increased in cross-modal settings, where modality-specific shifts further diverge the query and gallery distributions. A representative work, TCR (Li et al., 2025), addresses this by regulating modality uniformity and minimizing entropy on High-Confidence Queries. Distinct from these approaches that operate on raw embeddings and rely solely on positive certainty, our work complements them by explicitly rectifying structural textual noise and mining negative supervision from the typically discarded Low-Confidence Queries.

## 3. Method

In this section, we present ReEx, a unified framework designed to enhance the robustness of cross-modal TTA against structural noise and insufficient supervision. As illustrated in Fig. 2, our framework integrates two core modules into the online adaptation loop. First, explicitly targeting the textual modality, the Generative Semantic Refinement (GSR) module (Sec. 3.2) acts as a semantic anchor. It rectifies structurally corrupted text queries to ensure consistency before adaptation. With the textual features refined, Exclusion-Guided Proxy Contrastive Learning (EPCL) (Sec. 3.3) serves as the adaptation engine. By reformulating the learning objective, EPCL exploits the negative

certainty in ambiguous queries to refine decision boundaries, thereby maximizing the utilization of the query stream.

### 3.1. Problem Definition

We consider a cross-modal retrieval framework consisting of a query encoder $f_\Theta^Q$ and a gallery encoder $f_\Theta^G$. The model is pre-trained on a source dataset $\mathcal{D}_S$, which is inaccessible during inference. In the target domain, we are provided with a fixed gallery set $\mathcal{G} = \{g_i\}_{i=1}^{N_G}$ and a continuous stream of incoming queries $\mathcal{Q} = \{q_i\}_{i=1}^{N_Q}$. We assume a practical TTA setting where the query distribution shifts due to structural noise (*e.g.*, typos), while the gallery distribution remains relatively stable. The goal is to adapt a subset of parameters $\tilde{\Theta} \subseteq \Theta$ on-the-fly to maximize retrieval accuracy.

Following standard online TTA protocols (Li et al., 2025; Wang et al., 2021), adaptation is performed in a batch-wise manner. For an incoming query batch $\mathbf{x}_Q = \{q_i\}_{i=1}^B$, we extract normalized query embeddings $\mathbf{Z}_Q \in \mathbb{R}^{B \times D}$ and gallery embeddings $\mathbf{Z}_G \in \mathbb{R}^{N_G \times D}$. The retrieval prediction probability matrix $\mathbf{P} \in \mathbb{R}^{B \times N_G}$ is computed via softmax-normalized cosine similarity:

$$\mathbf{P}_{ij} = \frac{\exp(\mathbf{z}_{q,i} \cdot \mathbf{z}_{g,j}^\top / \tau)}{\sum_{k=1}^{N_G} \exp(\mathbf{z}_{q,i} \cdot \mathbf{z}_{g,k}^\top / \tau)}, \quad (1)$$

where $\tau$ is the temperature parameter. Conventional ap-

proaches (Wang et al., 2021; Liang et al., 2020) typically adapt the model by minimizing prediction entropy to enforce confident predictions:

$$\min_{\tilde{\Theta}} \mathcal{L}_{\text{ent}} = -\frac{1}{B} \sum_{i=1}^{B} \sum_{j=1}^{N_G} \mathbf{P}_{ij} \log \mathbf{P}_{ij}. \qquad (2)$$

To prevent error reinforcement, recent methods like TCR (Li et al., 2025) employ thresholding mechanisms to filter out Low-Confidence Queries. However, this strategy proves insufficient in the presence of structural noise. First, structural corruption often causes *miscalibration*, where the model assigns high confidence to incorrect predictions. This allows errors to bypass thresholds and degrade the adaptation process. Second, by treating low-confidence predictions merely as noise, these methods overlook informative *negative certainty*. This capability allows the model to identify definitively irrelevant candidates even when the exact match is ambiguous. Consequently, ignoring these signals wastes valuable supervision inherent in the query stream.

### 3.2. Generative Semantic Refinement

Conventional TTA approaches encode raw queries directly. While feasible for visual data, direct encoding of raw queries is suboptimal for text due to modal asymmetry. Unlike continuous, redundant visual data where encoders tolerate local noise, language is discrete; even minor typos disrupt tokenization and shift semantics. Thus, textual structural noise necessitates explicit generative rectification before embedding. To address this, we introduce the **Generative Semantic Refinement (GSR)** module. GSR utilizes the linguistic priors of lightweight Large Language Models (LLMs) to correct structural errors. However, relying solely on LLM generation introduces the risk of *semantic drift*, where the rectified output may alter the user's original intent despite being grammatically correct. To mitigate this, we employ a Confidence-Guided Dynamic Fusion (CGDF) mechanism that explicitly anchors the refinement to the original input cues, ensuring reliability.

**Structural Rectification.** We employ a lightweight LLM (*e.g.*, Qwen2.5-0.5B-Instruct (Yang et al., 2024a)) to rectify the raw batch $\mathbf{x}_Q = \{q_i\}_{i=1}^{B}$. Using a prompt $\mathcal{T}$ to correct typographical errors while preserving semantics, we obtain the refined batch $\mathbf{x}_{\hat{Q}} = \{\hat{q}_i\}_{i=1}^{B}$ as:

$$\mathbf{x}_{\hat{Q}} = \text{LLM}(\mathbf{x}_Q, \mathcal{T}). \qquad (3)$$

Note that this rectification is applied strictly to textual queries, as visual features exhibit greater invariance to low-level structural noise.

**Confidence-Guided Dynamic Fusion (CGDF).** Simply replacing the raw query $q_i$ with the rectified candidate $\hat{q}_i$ is risky, as the LLM might hallucinate or diverge from the

original intent. In contrast, the raw query, while noisy, retains the original token-level signals. To balance structural integrity and semantic fidelity, we propose a dynamic fusion mechanism. Assuming that alignment quality correlates with retrieval confidence, we assess the reliability of both raw and rectified queries based on their maximum prediction probabilities against the gallery:

$$c_i = \max(\mathbf{p}(q_i)), \quad \hat{c}_i = \max(\mathbf{p}(\hat{q}_i)), \qquad (4)$$

where $\mathbf{p}(\cdot)$ denotes the probability vector over the gallery (computed via Eq. 1). We then compute an adaptive weight $\alpha_i$ to prioritize the representation that better aligns with the target distribution:

$$\alpha_i = \frac{c_i}{c_i + \hat{c}_i}, \quad \hat{\alpha}_i = 1 - \alpha_i. \qquad (5)$$

Let $\mathbf{z}_i$ and $\hat{\mathbf{z}}_i$ denote the normalized embeddings of the raw and rectified queries, respectively. The final refined embedding $\tilde{\mathbf{z}}_i$ is obtained by weighted interpolation:

$$\tilde{\mathbf{z}}_i = \text{Normalize}\left(\alpha_i \mathbf{z}_i + \hat{\alpha}_i \hat{\mathbf{z}}_i\right). \qquad (6)$$

By dynamically adjusting $\alpha_i$, CGDF incorporates the LLM's correction only when it enhances alignment, thereby preventing semantic drift by reverting to the original cues when the generation is unreliable.

### 3.3. Exclusion-Guided Proxy Contrastive Learning

To effectively exploit supervision signals from all queries, we propose **Exclusion-Guided Proxy Contrastive Learning (EPCL)**. Unlike existing methods that discard Low-Confidence Queries during batch processing, EPCL processes all queries within each mini-batch and transforms their predictive ambiguity into informative exclusion-based constraints. The framework consists of three components: Semantic Sets Construction, Candidate-aware Proxy Alignment, and Exclusion-aware Negative Mining.

**Semantic Sets Construction.** For each refined query $q_i$ and the gallery set $\mathcal{G}$, we partition the prediction space into two subsets to identify potential positives and reliable negatives. First, we retrieve the top-$K$ nearest neighbors to form the Candidate Set $\mathcal{C}_i$, capturing the most likely semantic matches. Second, to construct the Exclusion Set $\mathcal{E}_i$, we adopt a batch-wise mining strategy. In online TTA, global negatives are often too distinct to provide useful gradients. To leverage the local batch context, we aggregate the candidate sets of all queries in the current batch into a shared pool $\mathcal{U} = \bigcup_{j=1}^{B} \mathcal{C}_j$. This pool represents the semantic subspace relevant to current batch data. From this pool, we identify the bottom-$K$ samples with the lowest similarity to $q_i$:

$$\mathcal{C}_i = \text{Top-}K(q_i, \mathcal{G}), \quad \mathcal{E}_i = \text{Bottom-}K(q_i, \mathcal{U}). \qquad (7)$$

By defining $\mathcal{E}_i$ as the least likely samples within the shared pool $\mathcal{U}$, we obtain hard negatives from the current distribution and establish an explicit semantic rejection boundary for $q_i$. Furthermore, we set $|\mathcal{E}_i| = |\mathcal{C}_i| = K$ to ensure balanced constraints and uniform computational overhead.

**Candidate-aware Proxy Alignment (CPA).** To optimize $q_i$, we construct a robust positive supervision signal based on $\mathcal{C}_i$. To mitigate the risk of noise in individual predictions, we formulate a **Soft Positive Proxy** $p_i^+$ by aggregating semantic information from the local neighborhood:

$$p_i^+ = \sum_{m \in \mathcal{C}_i} w_{i,m}\, \mathbf{z}_{g,m}, \quad w_{i,m} = \frac{\exp(\mathbf{z}_{q,i} \cdot \mathbf{z}_{g,m}^\top / \tau)}{\sum_{j \in \mathcal{C}_i} \exp(\mathbf{z}_{q,i} \cdot \mathbf{z}_{g,j}^\top / \tau)}. \tag{8}$$

This proxy mechanism exhibits inherent adaptivity. For High-Confidence Queries, the peaked probability distribution causes $p_i^+$ to converge toward a precise target. Conversely, for Low-Confidence Queries, the weights $w_{i,m}$ become uniform, resulting in a smoothed proxy that represents a broader semantic region rather than a specific, potentially incorrect point. This allows us to unify the treatment of High-Confidence and Low-Confidence Queries, learning precise discrimination from the former while preserving safe semantic context from the latter.

**Exclusion-aware Negative Mining (ENM).** To complement the positive proxy $\mathcal{P}_i = \{p_i^+\}$, we construct a negative set $\mathcal{N}_i$ that integrates constraints from both the gallery and the current query batch:

$$\mathcal{N}_i = \{\mathbf{z}_{g,m} \mid m \in \mathcal{E}_i\} \cup \{\mathbf{z}_{q,j} \mid \mathcal{C}_i \cap \mathcal{E}_j \neq \emptyset\}. \tag{9}$$

The first term utilizes explicit gallery negatives from the exclusion set $\mathcal{E}_i$. The second term mines **Implicit Negative Queries** from the current batch by exploiting semantic contradictions. Specifically, the condition $\mathcal{C}_i \cap \mathcal{E}_j \neq \emptyset$ implies that samples considered likely matches by $q_i$ are explicitly rejected by $q_j$. This contradiction indicates that $q_i$ and $q_j$ target divergent semantic regions. By treating such $q_j$ as a negative, we mine additional supervision from inter-query relationships, refining the embedding space beyond standard query-gallery alignment.

**Optimization Objective.** Having defined both positive proxies and negative samples, we now formulate a unified objective. By aligning each query with its adaptive proxy while contrasting against exclusion-based negatives, EPCL optimizes the following InfoNCE loss over the batch $B$:

$$\mathcal{L}_{\text{EPCL}} = -\frac{1}{|B|} \sum_{i \in B} \log \frac{\exp(\mathbf{z}_{q,i} \cdot p_i^+ / \tau)}{\sum_{\mathbf{z} \in \mathcal{P}_i \cup \mathcal{N}_i} \exp(\mathbf{z}_{q,i} \cdot \mathbf{z} / \tau)}. \tag{10}$$

This objective effectively pulls the query towards the estimated semantic center and pushes it away from all semantically incompatible samples, enabling robust learning from both High-Confidence and Low-Confidence Queries.

## 4. Experiments

### 4.1. Experiments Settings

**Datasets and Protocols.** Following established benchmarks (Li et al., 2025), we evaluate our method under two distinct settings. First, **Query Shift (QS)** simulates out-of-distribution queries while the gallery remains unchanged. We utilize **COCO-C** (Li et al., 2025) and **Flickr-C** (Li et al., 2025), derived from the test sets of COCO (Lin et al., 2014) and Flickr30k (Plummer et al., 2015). Second, **Query-Gallery Shift (QGS)** evaluates generalization where both query and gallery distributions diverge from the source. We benchmark on **Fashion-Gen** (Rostamzadeh et al., 2018) and **Nocaps** (Agrawal et al., 2019) alongside standard datasets to assess performance across diverse domains.

**Corruption Types.** To strictly evaluate robustness against structural noise, the benchmarks incorporate comprehensive perturbations across modalities. The image modality includes **16 Corruptions** categorized into four groups: *Noise* (Gaussian, Shot, Impulse, Speckle), *Blur* (Defocus, Glass, Motion, Zoom), *Weather* (Snow, Frost, Fog, Brightness), and *Digital* (Contrast, Elastic, Pixelate, JPEG). Simultaneously, the text modality features **15 Corruptions** spanning three linguistic levels: (1) *Character-level* perturbations include OCR errors, Insertion (CI), Replacement (CR), Swap (CS), and Deletion (CD); (2) *Word-level* shifts involve Synonym Replacement (SR), Random Insertion (RI), Swap (RS), Deletion (RD), and Punctuation Insertion (IP); and (3) *Sentence-level* styles span Formal, Casual, Passive, Active, and Back-translation.

**Backbones and Comparable Methods.** We adopt BLIP (Li et al., 2022) as the source model, owing to its strong performance and widespread use in image–text retrieval tasks. Consistent with (Li et al., 2025; Lee et al., 2018), we conduct evaluations for both image-to-text (I2T) and text-to-image (T2I) retrieval. To evaluate ReEx against existing TTA methods in cross-modal retrieval, we implement six comparable approaches using their publicly available open-source code: TENT (Wang et al., 2021), EATA (Niu et al., 2022), SAR (Niu et al., 2023), READ (Yang et al., 2024b), DeYO (Lee et al., 2024), TCR (Li et al., 2025).

**Implementation Details.** For the Query Shift setting, we adopt the backbone from the official weights, which have been fine-tuned on COCO and Flickr, and evaluate it on the corresponding corrupted datasets COCO-C or Flickr-C. For the Query-Gallery Shift setting, we use the official weights without any dataset-specific fine-tuning and conduct experiments on COCO, Flickr, and Nocaps to assess cross-domain generalization. During adaptation, ReEx optimizes Eqn. 10 with a batch size of 64. Following the setting in TCR (Li et al., 2025), we optimize 3 steps per batch, updating only the Layer Normalization (LN) layers of $f_\Theta^Q$.

*Table 1.* Comparison of Image-to-Text retrieval Recall@1 on COCO-C and Flickr-C under Query Shift. Best results are shown in **bold** .

| | Method | Noise | | | | Blur | | | | Weather | | | | Digital | | | | Avg. |
|---|---|---|---|---|---|---|---|---|---|---|---|---|---|---|---|---|---|---|
| | | Gauss. | Shot. | Impul. | Speckle | Defoc. | Glass | Motion | Zoom | Snow | Frost | Fog | Brit. | Contr. | Elastic | Pixel | JPEG | |
| COCO-C | w/o ADAPT | 43.4 | 46.3 | 43.2 | 57.3 | 43.3 | 68.0 | 39.7 | 8.4 | 32.3 | 52.2 | 57.0 | 66.8 | 36.0 | 41.3 | 20.6 | 63.7 | 45.0 |
| | TENT | 42.5 | 34.9 | 40.3 | 64.5 | 44.7 | 71.8 | 39.6 | 8.3 | 31.9 | 46.6 | 56.7 | 70.4 | 31.8 | 41.1 | 20.6 | 67.8 | 44.6 |
| | SHOT | 52.7 | 54.6 | 55.2 | 64.2 | 57.8 | 72.8 | 55.5 | 30.4 | 54.6 | 62.3 | 69.6 | 71.7 | 56.0 | 61.8 | 41.0 | 68.2 | 58.0 |
| | EATA | 44.5 | 48.1 | 48.2 | 61.9 | 37.5 | 72.5 | 43.0 | 7.3 | 36.5 | 55.4 | 62.1 | 70.2 | 38.3 | 43.5 | 17.7 | 65.7 | 47.0 |
| | SAR | 43.3 | 41.6 | 45.8 | 61.5 | 38.0 | 71.1 | 39.3 | 6.1 | 31.5 | 53.9 | 62.7 | 70.0 | 32.6 | 48.1 | 16.7 | 66.4 | 45.2 |
| | READ | 46.4 | 46.9 | 36.0 | 62.0 | 47.1 | 71.5 | 39.8 | 11.5 | 42.9 | 49.9 | 60.0 | 70.2 | 33.4 | 41.0 | 16.8 | 66.3 | 46.4 |
| | DeYO | 47.9 | 53.5 | 45.3 | 59.1 | 42.9 | 70.4 | 36.7 | 5.0 | 37.5 | 59.7 | 66.4 | 71.2 | 40.3 | 44.8 | 14.1 | 65.5 | 47.5 |
| | TCR | 51.5 | 55.5 | 54.8 | 61.0 | 56.3 | 72.7 | 57.1 | 32.2 | 55.5 | 63.5 | 70.9 | 72.0 | 59.5 | 61.9 | 42.6 | 68.4 | 58.5 |
| | **Ours** | **54.2** | **56.5** | **55.2** | 64.3 | **59.1** | **73.5** | **58.1** | **36.2** | **58.8** | **63.7** | **71.1** | **72.8** | **60.0** | **65.1** | **43.9** | **68.5** | **60.1** |
| Flickr-C | w/o ADAPT | 49.8 | 56.6 | 50.3 | 71.6 | 53.1 | 84.5 | 47.4 | 15.5 | 66.4 | 80.4 | 79.5 | 85.5 | 60.6 | 53.3 | 35.1 | 80.3 | 60.6 |
| | TENT | 55.0 | 63.4 | 56.4 | 76.4 | 39.0 | **88.0** | 23.5 | 1.6 | 72.9 | 85.3 | 84.2 | 88.8 | 71.7 | 63.1 | 35.4 | 82.6 | 61.7 |
| | SHOT | 61.0 | 65.6 | 62.2 | 79.0 | 70.2 | 87.9 | 65.0 | 38.4 | 78.1 | 85.2 | 84.7 | 88.7 | 73.8 | 72.9 | 56.4 | **83.3** | 72.0 |
| | EATA | 51.0 | 58.1 | 52.8 | 73.9 | 56.0 | 86.1 | 47.8 | 13.5 | 69.1 | 82.0 | 81.4 | 86.8 | 61.9 | 56.6 | 34.2 | 81.4 | 62.0 |
| | SAR | 54.3 | 62.5 | 55.6 | 75.2 | 48.4 | 87.3 | 34.8 | 15.5 | 72.0 | 83.3 | 82.1 | 87.9 | 68.4 | 60.6 | 42.2 | 81.4 | 63.2 |
| | READ | 50.1 | 58.2 | 52.3 | 74.8 | 63.7 | 87.0 | 55.1 | 2.2 | 71.7 | 83.7 | 82.1 | 87.4 | 67.4 | 62.3 | 42.5 | 81.4 | 63.9 |
| | DeYO | 54.7 | 63.8 | 56.3 | 76.6 | 63.8 | 86.8 | 50.3 | 3.2 | 74.8 | 83.7 | 82.1 | 88.7 | 65.7 | 63.1 | 46.8 | 83.2 | 65.2 |
| | TCR | 62.0 | 66.6 | 61.4 | 80.0 | 68.1 | 87.9 | 65.2 | 39.9 | 78.2 | 85.2 | 85.7 | 89.5 | 75.1 | 73.1 | 56.8 | **83.3** | 72.4 |
| | **Ours** | **64.9** | **68.4** | **66.4** | **81.9** | 68.4 | 87.5 | **69.9** | **50.4** | **79.7** | **85.6** | **86.0** | **89.7** | **77.2** | **77.6** | **61.3** | 82.2 | **74.8** |

We leverage Qwen2.5-0.5B-Instruct (Yang et al., 2024a) as a lightweight LLM, prompting it to reconstruct sentences into clear, fluent English while ensuring logical consistency and physical plausibility. For hyperparameters, we set $k = 3$ and $\tau = 0.1$, employing the AdamW optimizer (Loshchilov & Hutter, 2019) with learning rates of $8e^{-4}$ and $3e^{-5}$ for image and text retrieval, respectively. All experiments are conducted with two NVIDIA RTX 3090 GPUs.

## 4.2. Retrieval Results under Query Shift

In this section, we evaluate the robustness of different methods under the Query Shift setting, where only the query distribution is corrupted while the gallery remains unchanged. Results are reported on COCO-C and Flickr-C for both I2T and T2I tasks, as summarized in Table 1 and Table 2. ReEx consistently outperforms the strong baseline TCR, achieving improvements of 1.6% (I2T) and 1.1% (T2I) on COCO-C, and 2.4% (I2T) and 2.3% (T2I) on Flickr-C.

**Image-to-Text Retrieval.** For the image-to-text retrieval task, ReEx demonstrates consistent robustness across both the COCO-C and Flickr-C benchmarks. On COCO-C, ReEx surpasses TCR by a significant margin under severe degradations such as Zoom Blur (+4.0%), Snow (+3.3%), Elastic (+3.2%), and Pixelate compression (+1.3%). While these corruptions substantially degrade visual quality and often lead standard test-time adaptation methods to overfit noisy predictions, ReEx leverages EPCL to explicitly suppress unlikely candidates, thereby stabilizing the optimization process. This robustness is further evidenced on Flickr-C, where ReEx achieves a mean Recall@1 of 74.8%. The performance advantage is particularly notable under severe blur and digital noise, suggesting that the proposed approach effectively preserves cross-modal alignment even when visual information is significantly compromised.

**Text-to-Image Retrieval.** For the text-to-image retrieval task, ReEx similarly demonstrates superior performance across various textual perturbations. On the COCO-C dataset, the method improves upon TCR by 1.1%, raising the average Recall@1 from 41.7% to 42.8%. The gains are most pronounced under character-level corruptions, such as insertion (+4.7%), swapping (+3.4%), and deletion (+1.3%), as well as under word-level and sentence-level noise. Such perturbations often disrupt tokenization and semantic consistency, presenting a significant challenge for standard retrieval models. The robustness of ReEx in this domain is attributed to the Generative Semantic Refinement (GSR) module, which utilizes confidence-guided fusion to restore syntactic and semantic coherence while retaining original cues. On Flickr-C, the improvement over the TCR extends to 2.3%, with ReEx achieving an average Recall@1 of 65.7%. The method shows distinct advantages in handling character-level noise where conventional approaches struggle due to tokenization errors, highlighting the efficacy of semantic refinement in maintaining retrieval accuracy under linguistic distribution shifts.

## 4.3. Retrieval Results under Query-Gallery Shift

We further evaluate the robustness of different methods under the more challenging Query-Gallery Shift setting, where both queries and gallery samples deviate from the source distribution. Results are reported on four datasets with varying domain gaps, including Flickr, COCO, Fashion-Gen, and Nocaps.

As summarized in Table 3, ReEx achieves the highest average Recall@1 of 66.8%, surpassing the baseline by 4.7% and TCR by 0.4%. Specifically, the proposed method demonstrates strong generalization on general-domain datasets, reaching 79.6% on Flickr and 58.6% on

*Table 2.* Comparison of Text-to-Image retrieval Recall@1 on COCO-C and Flickr-C under Query Shift. Best results are shown in **bold**.

| Method | | Character-level | | | | | Word-level | | | | | Sentence-level | | | | | Avg. |
|---|---|---|---|---|---|---|---|---|---|---|---|---|---|---|---|---|---|
| | | OCR | CI | CR | CS | CD | SR | RI | RS | RD | IP | Formal | Casual | Passive | Active | Backtrans | |
| COCO-C | w/o ADAPT | 31.4 | 10.7 | 9.4 | 17.6 | 11.5 | 43.3 | 50.7 | 50.2 | 50.3 | 56.8 | 56.5 | 56.4 | 55.0 | 56.9 | 54.2 | 40.7 |
| | TENT | 30.9 | 2.5 | 2.0 | 7.3 | 3.6 | 44.1 | 52.3 | 51.2 | 50.6 | 57.2 | 56.6 | 56.5 | 55.6 | 57.2 | 54.0 | 38.8 |
| | SHOT | 32.5 | 11.7 | 10.2 | 18.2 | 12.0 | 43.9 | 51.2 | 50.7 | 50.8 | 57.3 | 57.0 | 56.8 | 55.5 | 57.4 | 54.5 | 41.3 |
| | EATA | 32.3 | 11.2 | 9.6 | 18.0 | 11.7 | 43.9 | 51.5 | 50.8 | 50.8 | 57.2 | 57.0 | 56.9 | 55.6 | 57.4 | 54.5 | 41.2 |
| | SAR | 31.9 | 11.3 | 9.6 | 18.5 | 12.0 | 43.3 | 50.7 | 50.2 | 50.3 | 56.8 | 56.4 | 56.4 | 55.0 | 56.9 | 54.2 | 40.9 |
| | READ | 32.0 | 11.1 | 10.1 | 17.5 | 11.5 | 44.1 | 52.3 | **51.4** | 50.9 | 57.4 | 56.9 | 56.8 | 56.0 | 57.5 | 54.6 | 41.3 |
| | DeYO | 32.4 | 11.2 | 9.6 | 18.0 | 11.8 | 44.0 | 51.7 | 50.9 | 51.0 | 57.4 | 57.2 | 57.0 | 55.7 | 57.5 | 54.7 | 41.3 |
| | TCR | 33.2 | 12.8 | 11.5 | 18.9 | 12.7 | 44.7 | 52.0 | 50.6 | 51.2 | 57.3 | 57.1 | 56.9 | 55.5 | 57.4 | 54.4 | 41.7 |
| | Ours | **35.9** | **17.5** | **12.3** | **22.3** | **14.0** | **44.8** | **52.8** | 51.1 | 51.1 | **57.6** | 57.1 | 56.8 | 56.1 | 57.5 | 54.7 | **42.8** |
| Flickr-C | w/o ADAPT | 53.5 | 18.4 | 18.0 | 30.4 | 22.5 | 68.3 | 77.9 | 76.9 | 77.9 | 82.1 | 82.1 | 81.9 | 79.9 | 82.2 | 79.8 | 62.1 |
| | TENT | 56.4 | 16.6 | 13.4 | 31.1 | 23.0 | 69.6 | 78.8 | 77.7 | 78.0 | **82.6** | 82.2 | 82.0 | 80.6 | 82.7 | 80.1 | 62.3 |
| | SHOT | 57.0 | 21.4 | 21.7 | 32.9 | 24.4 | 69.6 | 78.6 | 77.9 | 78.1 | 82.2 | 82.0 | 81.9 | 80.6 | 82.5 | 80.1 | 63.4 |
| | EATA | 55.7 | 19.2 | 19.1 | 31.5 | 23.4 | 69.1 | 78.5 | 77.7 | 78.0 | 82.5 | 82.2 | 82.0 | 80.4 | 82.4 | 80.1 | 62.8 |
| | SAR | 53.5 | 20.1 | 19.1 | 32.1 | 23.8 | 68.3 | 77.9 | 76.9 | 77.9 | 82.1 | 82.1 | 81.9 | 79.9 | 82.2 | 79.8 | 62.5 |
| | READ | 55.8 | 19.7 | 20.6 | 32.0 | 23.5 | 69.3 | 78.6 | 77.6 | 78.1 | 82.4 | 82.2 | 81.8 | 80.5 | 82.6 | 80.2 | 63.0 |
| | DeYO | 56.4 | 19.8 | 13.5 | 32.0 | 23.6 | 69.5 | 78.8 | 77.8 | 78.0 | 82.4 | 82.3 | 82.0 | 80.8 | 82.4 | 80.0 | 62.6 |
| | TCR | 56.9 | 21.1 | 21.5 | 32.7 | 24.1 | 69.6 | 78.7 | 77.7 | 78.1 | 82.5 | 82.2 | 82.1 | 80.7 | 82.5 | 80.4 | 63.4 |
| | Ours | **60.6** | **33.4** | **25.2** | **41.7** | **27.8** | **70.2** | **79.5** | 77.9 | 77.9 | 82.5 | **82.4** | 81.9 | 80.9 | 82.6 | 80.4 | **65.7** |

*Table 3.* Comparison of average Recall@1 for both Text-to-Image and Image-to-Text retrieval under Query-Gallery Shift across multiple domain datasets. In the table, "ID", "ND" and "OD" refer to "In-Domain", "Near-Domain" and "Out-Domain", respectively. Best results are shown in **bold**.

| Method | Flickr | COCO | Fashion | Nocaps | | | Avg. |
|---|---|---|---|---|---|---|---|
| | | | | ID | ND | OD | |
| w/o ADAPT | 69.2 | 52.4 | 23.0 | 81.6 | 71.5 | 74.9 | 62.1 |
| TENT | 75.2 | 51.7 | 20.1 | 82.0 | 73.4 | 75.8 | 63.0 |
| EATA | 75.9 | 56.1 | 19.0 | 81.5 | 73.4 | 74.7 | 63.4 |
| SAR | 75.0 | 55.1 | 22.0 | 81.9 | 73.2 | 75.3 | 63.8 |
| READ | 75.0 | 54.3 | 14.9 | 81.2 | 72.3 | 74.3 | 62.0 |
| DeYO | 76.7 | 56.2 | 18.2 | 82.4 | 74.7 | 76.9 | 64.2 |
| TCR | 78.1 | 57.8 | **25.7** | 82.5 | **76.3** | **78.2** | 66.4 |
| Ours | **79.6** | **58.6** | **25.7** | **82.7** | 76.1 | 77.9 | **66.8** |

COCO. In specialized domains, ReEx proves robust, matching the top performance on the fine-grained Fashion-Gen benchmark at 25.7% and maintaining competitive results across the In-Domain (82.7%), Near-Domain (76.1%), and Out-Domain (77.9%) splits of Nocaps. This consistent superiority under joint distribution shifts is attributed to the synergistic effect of GSR, which mitigates semantic drift in textual representations, and EPCL, which stabilizes adaptation by leveraging negative evidence when reliable positive pairs are scarce.

ReEx is primarily designed for query-side corruption. Therefore, its advantage is more pronounced in the QS setting, where the gallery remains stable and GSR can directly repair corrupted textual queries before EPCL performs adaptation. In QGS, however, the shift occurs on both the query and gallery sides, and the bottleneck becomes cross-domain feature misalignment rather than query corruption alone. As

a result, query-side semantic refinement contributes less in this setting, and the EPCL can become less reliable. This explains why ReEx still improves over existing TTA baselines under QGS, but with a smaller margin than in QS.

### 4.4. Computational Complexity Analysis of ReEx

*Table 4.* Comparison of retrieval performance and computational efficiency on Flickr-C dataset. Best results are shown in **bold**.

| Method | I2T | | T2I | | Avg. R@1 |
|---|---|---|---|---|---|
| | Recall@1 | Runtime | Recall@1 | Runtime | |
| w/o ADAPT | 60.6 | - | 62.1 | - | 61.4 |
| TENT | 61.7 | 2.00 | 62.3 | 0.23 | 62.0 |
| SHOT | 72.0 | 2.02 | 63.4 | 0.24 | 67.7 |
| EATA | 62.0 | **1.95** | 62.8 | 0.23 | 62.4 |
| SAR | 63.2 | 4.02 | 62.5 | 0.43 | 62.9 |
| READ | 63.9 | 2.02 | 63.0 | 0.23 | 63.4 |
| DeYO | 65.2 | 2.01 | 62.6 | **0.22** | 63.9 |
| TCR | 72.4 | 2.01 | 63.4 | 0.24 | 67.9 |
| Ours | **74.8** | 2.01 | **65.7** | 0.75 | **70.2** |

Table 4 compares the average Recall@1 and runtime per batch on Flickr-C under both I2T and T2I settings. We report the average Recall@1 across all corruptions and the runtime ($s$) per batch with a batch size of 64. In the I2T setting, ReEx achieves the highest average retrieval accuracy of 74.8%, outperforming the second-best TCR method by 2.4%, while maintaining a comparable runtime (2.01$s$). In the T2I setting, although incorporating LLM introduces an additional 0.5$s$ per batch, ReEx achieves an average retrieval accuracy of 65.7%, surpassing TCR by 2.3%. Although TCR improves on the baseline (62.1%) by only 1.3%, ReEx achieves a substantial gain of 3.6%, reaching 65.7%. This indicates that our method delivers a

favorable trade-off between accuracy and efficiency under structural text corruption. In practical retrieval systems, the average latency can be further reduced by confidence-gated routing, where high-confidence queries bypass GSR and only low-confidence queries are refined by the LLM.

## 4.5. Additional Experiments on BLIP-2

To further examine whether ReEx transfers to stronger VLP backbones, we additionally evaluate it on BLIP-2 (Li et al., 2023) backbone. As shown in Table 5, ReEx achieves the best average R@1 and improves over the strongest TTA baseline TCR by $1.5\%$. These results suggest that the proposed adaptation mechanism is not restricted to the BLIP backbone and can also benefit a stronger VLP model.

*Table 5.* Comparison of average Recall@1 for both Image-to-Text and Text-to-Image retrieval under Query Shift using the BLIP-2. Best results are shown in **bold**.

| Method | I2T | T2I | Avg. |
|---|---|---|---|
| w/o ADAPT | 65.0 | 56.9 | 61.0 |
| TENT | 64.8 | 56.8 | 60.8 |
| SHOT | 72.4 | 58.3 | 65.4 |
| EATA | 65.0 | 57.4 | 61.2 |
| SAR | 64.9 | 56.9 | 60.9 |
| READ | 64.9 | 57.7 | 61.3 |
| DeYO | 64.9 | 57.4 | 61.2 |
| TCR | 72.0 | 59.1 | 65.6 |
| Ours | **73.1** | **61.0** | **67.1** |

## 4.6. Ablation Study

This subsection evaluates the contribution of each component in the proposed framework. We conduct a comprehensive ablation study on the Flickr-C dataset. All ablation experiments are reported under the text-to-image retrieval setting. Table 6 reports the quantitative results.

*Table 6.* Ablation study on Flickr-C. We report Text-to-Image Recall@1 under the text-to-image retrieval setting. Best results are shown in **bold**.

| GSR | | EPCL | | Avg. |
|---|---|---|---|---|
| LLM | CGDF | CPA | ENM | |
| × | × | × | × | 62.1 |
| ✓ | × | × | × | 62.5 |
| × | × | ✓ | ✓ | 63.3 |
| ✓ | × | ✓ | ✓ | 63.2 |
| ✓ | ✓ | × | ✓ | 65.5 |
| ✓ | ✓ | ✓ | × | 64.3 |
| ✓ | ✓ | ✓ | ✓ | **65.7** |

**Effectiveness of components in GSR.** We remove the CGDF module and denote this variant as ReEx (w/o CGDF). In this setting, the LLM-corrected text directly replaces the

original query. As shown in Table 6, ReEx (w/o CGDF) achieves $63.2\%$, falling behind the original noisy baseline ($63.3\%$). This reveals a critical insight: blindly applying LLM corrections can be detrimental. Without the regulation of CGDF, the LLM tends to introduce semantic drift that hurts retrieval accuracy even more than the original structural noise. This confirms that our fusion mechanism is essential to prevent such "over-repairing" and ensure the model relies on the LLM only when it is trustworthy. In comparison, the full ReEx improves performance to $65.7\%$, yielding a $2.5\%$ gain over ReEx (w/o CGDF).

**Effectiveness of components in EPCL.** We next analyze the role of each component in EPCL. We remove the ENM module and update the model using only CPA, denoted as ReEx (w/o ENM). The full model outperforms this variant by $1.4\%$, highlighting the importance of mutual exclusivity in complementing CPA when handling low-confidence queries. Finally, we replace CPA with hard pseudo-labels ($k = 1$), denoted as ReEx (w/o CPA). This leads to a modest $0.2\%$ performance drop, indicating that CPA serves as a domain-agnostic complement that helps smooth neighborhood variance and maintain stable adaptation.

**Sensitivity Analysis.** We evaluate the model's sensitivity to hyperparameters by setting the neighborhood size $K \in 1, 3, 5, 7, 9$ and varying the temperature $\tau$ on a logarithmic scale from $10^{-4}$ to $1$. As illustrated in Fig. 3a, accuracy improves as $K$ increases from 1 to 3, indicating that aggregating information from a small local neighborhood effectively reduces prediction noise. Further increasing $K$ leads to a slight performance drop, likely due to the inclusion of semantically irrelevant samples. The small variance of 0.003 indicates that the model is not sensitive to the choice of $K$. Fig. 3b shows that performance peaks at $\tau = 0.1$. Small values of $\tau$ lead to overly sharp distributions that approximate hard labels, while larger values overly smooth the distribution and reduce discriminative ability. Despite this behavior, the variance of 0.116 indicates that the model maintains relatively stable performance across the evaluated range of $\tau$.

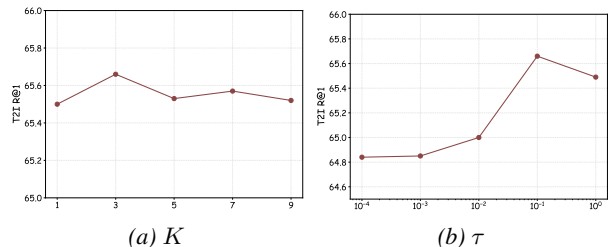

*(a)* $K$      *(b)* $\tau$

*Figure 3.* Sensitivity analysis of hyperparameters.

**Analysis of LLM Choices.** To investigate the impact of different LLM scales on ReEx performance and runtime, we compare SmolLM2-360M-Instruct (Allal et al., 2025),

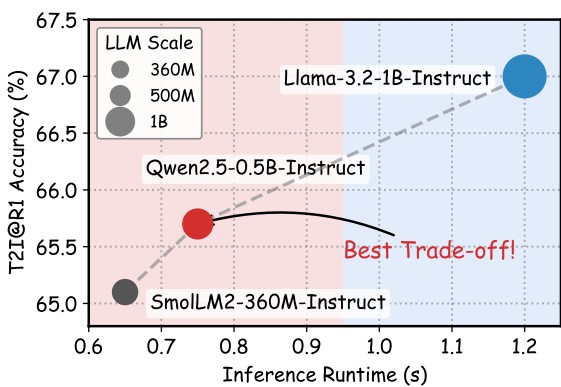

*Figure 4.* Performance-Efficiency trade-off with LLM scales.

Qwen2.5-0.5B-Instruct (Yang et al., 2024a), and Llama-3.2-1B-Instruct (Meta AI, 2024) in terms of accuracy and inference latency in Fig. 4. Larger models achieve higher accuracy due to stronger semantic understanding, but incur higher latency. Llama-3.2-1B-Instruct achieves the highest accuracy of $67.0\%$ with 1.6 times longer inference time than Qwen2.5-0.5B-Instruct while SmolLM2-360M-Instruct obtains lower accuracy of $65.1\%$ due to limited model capacity. Overall, Qwen2.5-0.5B-Instruct offers the best trade-off between accuracy and efficiency.

## 5. Conclusion and Limitations

We propose ReEx, a robust TTA framework addressing VLP vulnerability to real-world noise. By integrating Generative Semantic Refinement (GSR) to rectify textual structure and Exclusion-Guided Proxy Contrastive Learning (EPCL) to mine supervision from Low-Confidence Queries, ReEx effectively mitigates the confirmation bias of prior methods. Extensive experiments confirm that ReEx achieves state-of-the-art performance on COCO-C and Flickr-C.

The primary limitation of ReEx is the additional latency introduced by the LLM-based GSR module ($0.75s$ vs. $0.24s$ per batch). In this work, we apply GSR to all textual queries to evaluate robustness under severe corruptions, while in practical deployments it can serve as a conditional fallback module that is triggered only for Low-Confidence Queries to reduce amortized latency. Moreover, ReEx is most effective when the dominant shift occurs on the query side. Under Query-Gallery Shift, where both query and gallery distributions may change, the main challenge becomes broader cross-domain feature misalignment, which makes query-side refinement less effective and weakens the reliability of candidate/exclusion sets in EPCL. Finally, GSR currently focuses on textual queries, while visual corruptions involve diverse degradation types such as noise, blur, weather, and digital artifacts. Future work will explore lightweight correction models, knowledge distillation, and general visual refinement to improve efficiency and robustness.

## Acknowledgments

This work was supported in part by the National Natural Science Foundation of China under Grants 62506280, U22A2096 and 62576261, in part by the China Postdoctoral Science Foundation under Grant Number 2025M771559, in part by the Postdoctoral Fellowship Program of CPSF under Grant Number GZB20250399, in part by Scientific and Technological Innovation Teams in Shaanxi Province under grant 2025RS-CXTD-011, in part by the Shaanxi Province Core Technology Research and Development Project under grant 2024QY2-GJHX-11, and in part by the CCF-Kuaishou Large Model Explorer Fund (NO. CCF-KuaiShou 2025006).

## Impact Statement

This paper presents work whose goal is to advance the field of machine learning. There are many potential societal consequences of our work, none of which we feel must be specifically highlighted here.

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

## A. Prompts for Generative Semantic Refinement

This section details the prompt templates used in our GSR module. We design a specialized instruction set to guide the LLM in restoring corrupted captions. The complete prompt structure and generation hyperparameters are provided in Box A.

---

**Prompt for GSR**

**1. Prompt Template**
We construct the input using the standard chat template.
**[System Message]:**
```
You are a helpful assistant for text refinement.  The input text is a noisy
description of a visual scene.  Reconstruct the sentence into clear, fluent English.
Ensure the description is logically consistent and describes a physically plausible
scene.  Output only the refined text without punctuation.
```
**[User Message]:**
```
Input:  {Noisy_Input_Text}
Output:
```
---

**2. Generation Configuration**
To ensure deterministic and stable outputs, we utilize the following sampling parameters:
**Temperature:** 0    **Max New Tokens:** 64    **Precision:** FP16    **Stop Tokens:** Model-specific EOS tokens

---

## B. Additional Visualization Results

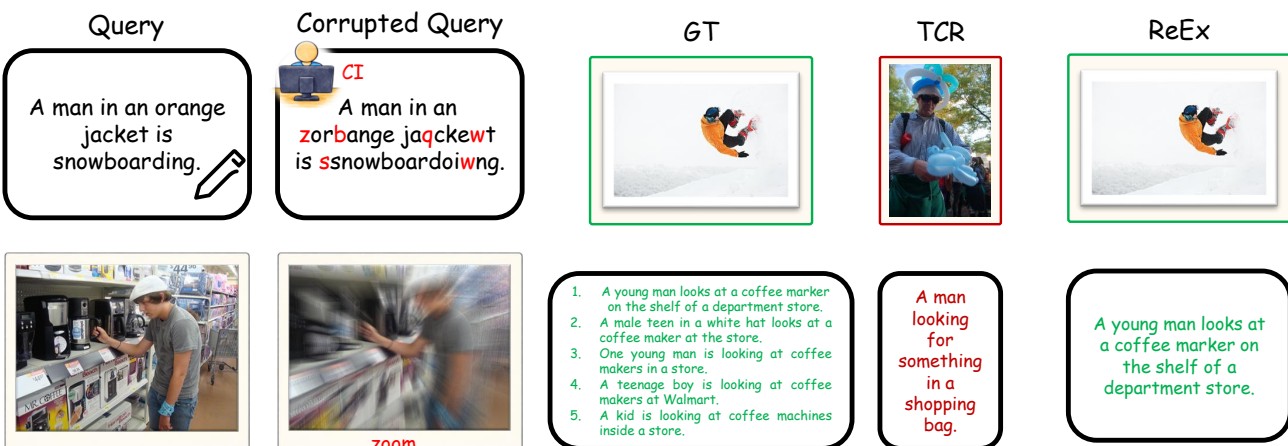

*Figure 5.* Qualitative comparison of retrieval robustness under corruption. The top row illustrates a Text-to-Image case subject to Character Injection (CI), while the bottom row displays an Image-to-Text case degraded by Zoom Blur. In both instances, the baseline TCR fails to retrieve the correct target due to noise-induced misalignment. In contrast, ReEx successfully retrieves the ground truth.

Fig. 5 visualizes the retrieval results of TCR and ReEx under Character Injection (CI) and Zoom Blur perturbations. The top row illustrates a Text-to-Image task where the input text is corrupted by CI. TCR retrieves an incorrect image due to the lack of a semantic correction mechanism. In contrast, ReEx employs the GSR module to rectify the semantic content, successfully retrieving the matching sample. The bottom row displays an Image-to-Text case affected by Zoom Blur. TCR yields incorrect results as it fails to effectively utilize supervision signals from low-confidence queries. Conversely, ReEx achieves accurate retrieval by leveraging EPCL to impose constraints on negative samples, thereby enhancing model robustness.

## C. Detailed Experimental Results

### C.1. Detailed Results under Query-Gallery Shift

Due to space limitations in the main paper, Table 3 reports only the average Recall@1 across text-to-image and image-to-text for the QGS setting. In Table 7, we provide a detailed performance breakdown for each retrieval direction separately.

*Table 7.* Comparison of Recall@1 for both Text-to-Image and Image-to-Text retrieval under Query-Gallery Shift across multiple domain datasets. In the table, "ID", "ND" and "OD" refer to "In-Domain", "Near-Domain" and "Out-Domain", respectively. Best results are shown in **bold** .

| Method | Flickr | | COCO | | Fashion | | Nocaps(ID) | | Nocaps(ND) | | Nocaps(OD) | | Avg. |
| --- | --- | --- | --- | --- | --- | --- | --- | --- | --- | --- | --- | --- | --- |
| | I2T | T2I | I2T | T2I | I2T | T2I | I2T | T2I | I2T | T2I | I2T | T2I | |
| w/o ADAPT | 70.0 | 68.3 | 59.3 | 45.4 | 19.9 | 26.1 | 88.2 | 74.9 | 79.3 | 63.6 | 81.9 | 67.8 | 62.1 |
| TENT | 81.9 | 68.5 | 61.7 | 41.7 | 14.1 | 26.1 | 88.5 | 75.4 | 82.6 | 64.1 | 82.7 | 68.9 | 63.0 |
| EATA | 82.3 | 69.4 | 64.2 | 47.9 | 12.8 | 25.2 | 87.8 | 75.1 | 82.8 | 63.9 | 81.5 | 67.9 | 63.4 |
| SAR | 81.7 | 68.3 | 63.5 | 46.6 | 17.9 | 26.1 | 88.2 | 75.6 | 81.0 | 65.4 | 81.2 | 69.3 | 63.7 |
| READ | 80.0 | 69.9 | 62.1 | 46.4 | 5.6 | 24.1 | 87.3 | 75.1 | 80.6 | 63.9 | 80.7 | 67.9 | 62.0 |
| DeYO | 83.5 | 69.9 | 65.0 | 47.3 | 12.2 | 24.1 | 89.2 | 75.6 | 83.7 | 65.7 | 84.3 | 69.4 | 64.2 |
| TCR | 85.9 | 70.3 | 66.7 | **48.9** | 21.1 | **30.3** | 89.0 | **76.0** | 86.5 | **66.1** | **86.8** | **69.5** | 66.4 |
| Ours | **88.7** | **70.4** | **68.5** | 48.6 | **22.1** | 29.4 | **90.2** | 75.2 | **86.8** | 65.5 | **86.8** | 69.1 | **66.8** |

As shown in Table 7, the results show a significant advantage in the Image-to-Text setting, where ReEx consistently outperforms the strongest baseline TCR. Specifically, ReEx achieves 88.7% on Flickr (vs. 85.9% for TCR) and 68.5% on COCO (vs. 66.7% for TCR), corresponding to gains of +2.8% and +1.8%, respectively. In the Text-to-Image setting, ReEx maintains competitive performance, reaching 70.4% on Flickr compared to TCR's 70.3%. On the specialized Fashion-Gen dataset, ReEx achieves the highest I2T accuracy of 22.1%, surpassing TCR (21.1%) and significantly outperforming methods like READ (5.6%). Furthermore, on the Nocaps benchmark, ReEx leads in I2T accuracy across all splits, outperforming TCR by +1.2% on the In-Domain split (90.2% vs. 89.0%) and +0.3% on the Near-Domain split (86.8% vs. 86.5%).

## C.2. Per Corruption Results of Ablation Study

In Sec. 4.6, we evaluate the effectiveness of each component in ReEx through ablation studies. Table 8 further reports a detailed breakdown of performance across individual corruption types.

*Table 8.* Ablation study of Text-to-Image retrieval Recall@1 on Flickr-C under Query Shift. Best results are shown in **bold** .

| Method | Character-level | | | | | Word-level | | | | | Sentence-level | | | | | Avg. |
| --- | --- | --- | --- | --- | --- | --- | --- | --- | --- | --- | --- | --- | --- | --- | --- | --- |
| | OCR | CI | CR | CS | CD | SR | RI | RS | RD | IP | Formal | Casual | Passive | Active | Backtrans | |
| w/o ADAPT | 53.5 | 18.4 | 18.0 | 30.4 | 22.5 | 68.3 | 77.9 | 76.9 | 77.9 | 82.1 | 82.1 | 81.9 | 79.9 | 82.2 | 79.8 | 62.1 |
| only LLM | 55.9 | 29.6 | 20.0 | 35.8 | 22.3 | 67.1 | 76.4 | 72.7 | 75.0 | 81.5 | 81.2 | 80.7 | 79.2 | 81.2 | 79.1 | 62.5 |
| w/o LLM | 56.6 | 20.9 | 21.8 | 32.8 | 24.2 | 69.1 | 78.8 | 77.6 | **78.2** | 82.4 | 82.3 | **81.9** | 80.6 | 82.5 | 80.2 | 63.3 |
| w/o CGDF | 56.6 | 30.3 | 20.8 | 36.6 | 23.3 | 67.7 | 77.8 | 73.4 | 75.4 | 81.9 | 81.6 | 81.1 | 79.8 | 81.7 | 79.8 | 63.2 |
| w/o CPA | 60.6 | 33.3 | 25.1 | 41.6 | 27.6 | 70.1 | **79.5** | **77.9** | 77.7 | 82.3 | 82.1 | 81.7 | 80.7 | 82.4 | 80.3 | 65.5 |
| w/o ENM | 58.8 | 31.4 | 23.0 | 39.3 | 26.1 | 68.8 | 78.5 | 76.5 | 77.2 | 81.8 | 81.4 | 80.9 | 80.1 | 81.7 | 79.3 | 64.3 |
| Ours | **60.6** | **33.4** | **25.2** | **41.7** | **27.8** | 70.2 | **79.5** | **77.9** | 77.9 | **82.5** | **82.4** | **81.9** | **80.9** | **82.6** | **80.4** | **65.7** |

Removing the LLM leads to a substantial decline in average retrieval accuracy under character-level corruptions, dropping from 37.7% to 31.3%, indicating that when textual structures are severely disrupted, the LLM serves as a crucial semantic corrector for recovering lost semantic information. The w/o CGDF variant, which directly uses LLM-generated corrections for contrastive learning, achieves notable improvements over w/o LLM on character-level corruptions (e.g., CI improves from 20.9% to 30.3%), but exhibits clear performance degradation under word-level corruptions, where accuracy drops to 73.4% compared to 77.9% for the full model. This degradation confirms that directly relying on LLM outputs without CGDF introduces semantic drift, disrupting original word order and semantics. Removing CPA (w/o CPA) results in slightly lower performance than the full model across most metrics (e.g., Casual: 81.7% vs. 81.9%; CI: 33.3% vs. 33.4%); although the gaps are small, their consistent presence suggests that CPA, while not tailored to specific corruption types, contributes to the overall stability of the adaptation process. Finally, removing the ENM module (w/o ENM) causes a sustained performance drop under character-level corruptions (e.g., CI decreases to 31.4%), highlighting the importance of the exclusion mechanism in handling high-noise, low-confidence queries.

### C.3. Impact of LLM Backbone scale

To provide a more detailed analysis of the performance-efficiency trade-off illustrated in Fig. 4 of the main paper, Table 9 reports the numerical performance and inference latency of different large language model backbones.

*Table 9.* Quantitative comparison of LLM backbones. Best results are shown in **bold**.

| Method | Character-level | | | | | Word-level | | | | | Sentence-level | | | | | Avg. | Time |
|---|---|---|---|---|---|---|---|---|---|---|---|---|---|---|---|---|---|
| | OCR | CI | CR | CS | CD | SR | RI | RS | RD | IP | Formal | Casual | Passive | Active | Backtrans | | |
| w/o ADAPT | 53.5 | 18.4 | 18.0 | 30.4 | 22.5 | 68.3 | 77.9 | 76.9 | 77.9 | 82.1 | 82.1 | 81.9 | 79.9 | 82.2 | 79.8 | 62.1 | - |
| SmolLM2-360M | 60.2 | 31.4 | 24.6 | 39.8 | 27.3 | **70.2** | 78.9 | 77.6 | 77.2 | 82.4 | **82.5** | 81.7 | 80.8 | 82.4 | **80.4** | 65.1 | **0.65** |
| Qwen2.5-0.5B | 60.6 | 33.4 | 25.2 | 41.7 | 27.8 | **70.2** | 79.5 | 77.9 | **77.9** | **82.5** | 82.4 | **81.9** | **80.9** | **82.6** | **80.4** | 65.7 | 0.75 |
| Llama-3.2-1B | **63.3** | **42.3** | **27.8** | **48.0** | **30.0** | 70.0 | **80.1** | 77.3 | 77.3 | **82.5** | 82.3 | 81.8 | 80.6 | 82.2 | 79.9 | **67.0** | 1.20 |

Table 9 presents the detailed performance of different LLM backbones across three corruption types. We observe that the performance gap between models is mainly concentrated in Character-level. Llama-3.2-1B achieves the best results on corruptions like OCR, CI, and CS, indicating that a larger model size is helpful for correcting character errors. However, for Word-level and Sentence-level corruption, the performance difference becomes negligible. Smaller models like SmolLM2-360M and Qwen2.5-0.5B achieve similar or even slightly better results than Llama-3.2-1B on corruptions such as SR and Formal. Although Llama-3.2-1B achieves the highest average accuracy ($67.0\%$), it requires significantly more inference time ($1.20s$). Therefore, we select Qwen2.5-0.5B as the backbone because it offers the best trade-off, providing competitive accuracy ($65.7\%$) with a much lower latency ($0.75s$) compared to the 1B model.

## D. Comparable Methods in Experiments

In Sec. 4.1, we evaluate ReEx against various TTA methods on the COCO-C and Flickr-C datasets. The comparison includes TTA methods such as TENT (Wang et al., 2021), SHOT (Liang et al., 2020), EATA (Niu et al., 2022), SAR (Niu et al., 2023), READ (Yang et al., 2024b), DeYO (Lee et al., 2024) and TCR (Li et al., 2025). Below, we provide a detailed overview of these methods.

Test-time Entropy Minimization (TENT) (Wang et al., 2021) adapts the model during inference by minimizing the Shannon entropy of predictions on unlabeled test batches. For each incoming batch, the method first estimates the normalization statistics (mean and variance) directly from the data and then updates the channel-wise affine transformation parameters ($\gamma$ and $\beta$) in the normalization layers using gradients derived from the entropy loss, while keeping all other model weights frozen. The official repository is available at `https://github.com/DequanWang/tent`.

Source Hypothesis Transfer (SHOT) (Liang et al., 2020) addresses unsupervised domain adaptation without accessing source data by freezing the pre-trained source classifier and optimizing the feature encoder. The method aligns target features to the source hypothesis by jointly minimizing the entropy of predictions to ensure certainty and maximizing the diversity of the output distribution to avoid trivial solutions, while further refining the alignment using self-supervised pseudo-labels derived from class-wise centroids in the feature space. The official repository is available at `https://github.com/tim-learn/SHOT`.

Efficient Anti-forgetting Test-time Adaptation (EATA) (Niu et al., 2022) enhances test-time adaptation efficiency and prevents catastrophic forgetting by introducing an active sample selection strategy and a Fisher-based weight regularizer. The method selectively updates model weights using only reliable and non-redundant samples identified via entropy and diversity criteria while constraining important parameters from drastic changes using Fisher information estimated from pseudo-labeled test data. The official repository is available at `https://github.com/mr-eggplant/EATA`.

Sharpness-Aware and Reliable entropy minimization (SAR) (Niu et al., 2023) stabilizes test-time adaptation in dynamic wild scenarios by addressing model collapse often observed in batch-agnostic normalization layers. The method filters out high-entropy samples to eliminate noisy gradients and subsequently optimizes the remaining reliable samples by jointly minimizing prediction entropy and the sharpness of the loss landscape to encourage convergence to a flat minimum that is robust to noisy updates. The official repository is available at `https://github.com/mr-eggplant/SAR`.

REliable fusion and robust ADaptation (READ) (Yang et al., 2024b) addresses multi-modal reliability bias by employing a

self-adaptive attention mechanism to dynamically modulate cross-modal fusion. Additionally, it utilizes a confidence-aware objective function to amplify high-confidence predictions while mitigating the impact of noisy ones. The official repository is available at `https://github.com/XLearning-SCU/2024-ICLR-READ`.

Destroy Your Object (DeYO) (Lee et al., 2024) addresses the limitations of entropy-based test-time adaptation by revealing its inability to distinguish between reliable shape features and spurious correlations. The method introduces a Pseudo-Label Probability Difference (PLPD) metric to quantify the model's reliance on object structure by measuring prediction changes after applying destructive transformations like patch shuffling. By combining PLPD with entropy for sample selection and weighting, DeYO ensures the model adapts using only robust samples that depend on intrinsic object shapes rather than background noise. The official repository is available at `https://github.com/Jhyun17/DeYO`.

Test-time adaptation for Cross-modal Retrieval (TCR) (Li et al., 2025) addresses the query shift problem by employing a prediction refinement module to filter retrieval noise, alongside a joint objective function that enhances intra-modality uniformity and rectifies inter-modality gaps to preserve the common space. The official repository is available at `https://github.com/XLearning-SCU/2025-ICLR-TCR`.

