# OpenReview forum: "Robust Cross-Modal Retrieval via Generative Semantic Refinement and Exclusion-Guided Adaptation"
_ICML.cc/2026/Conference — ICML 2026 regular_

### Official Review · Reviewer_XkQs · 2026-03-06

**Soundness:** 3
**Presentation:** 4
**Significance:** 3
**Originality:** 3
**Overall Recommendation:** 4
**Confidence:** 4

**Summary:**

The paper introduces ReEx, a test-time adaptation framework for cross-modal retrieval designed to improve robustness under query distribution shifts, such as structural textual noise and visual corruptions. The method consists of two main components. First, a Generative Semantic Refinement (GSR) module employs a lightweight LLM together with a confidence-guided dynamic fusion mechanism to repair noisy text queries while reducing semantic drift. Second, an Exclusion-Guided Proxy Contrastive Learning (EPCL) module leverages low-confidence queries to construct exclusion sets of unlikely candidates. By exploiting ambiguous queries to mine informative hard negatives instead of discarding them, the method refines the retrieval embedding space during adaptation. Experiments on COCO-C, Flickr-C, and several cross-domain datasets show consistent improvements over existing test-time adaptation approaches, including TCR.

**Compliance With Llm Reviewing Policy:**

Affirmed.

**Final Justification:**

I thank the authors for the discussion throughout the discussion process; it helped clarify understanding certain points that I missed earlier. Although the authors addressed some of my questions regarding the framing and method detail of the paper, I think the paper can benefit from our discussion in the paper; resulting in retaining my positive score while increasing my understanding confidence in the paper.

**Key Questions For Authors:**

1. The GSR module uses an LLM to repair noisy queries. How does the method prevent semantic drift when the corruption is severe and the intended meaning of the query becomes ambiguous?
2. Since GSR involves invoking an LLM, could the authors clarify the additional inference latency and its impact on real-world retrieval systems?
3. The method updates only LayerNorm parameters during test-time adaptation. How sensitive is performance to the number of adaptation steps, and does the model risk drifting when many noisy queries appear consecutively?
4. Why was generative refinement considered exclusively for text? Given the severe drops observed under visual corruptions like Zoom Blur, would a lightweight visual refinement module yield similar synergistic benefits for the I2T task?

**Limitations:**

yes

**Strengths And Weaknesses:**

Strengths:
1. The integration of a lightweight LLM for online textual correction is a practical approach to handling discrete language noise, and the Confidence-Guided Dynamic Fusion (CGDF) mechanism effectively mitigates the risk of semantic drift caused by LLM hallucinations.
2. The conceptualization of Exclusion-Guided Proxy Contrastive Learning (EPCL) is insightful. Repurposing typically discarded low-confidence queries into a source of negative constraints is a smart way to maximize the utility of the test stream and avoid confirmation bias.
3. The paper is well-written and logically organized. Figure 2 clearly and effectively visualizes the dual-module architecture, making the interplay between generative refinement and proxy contrastive learning easy to grasp.
4. The experimental protocol is well-designed to evaluate robustness under distribution shift. In particular, the paper evaluates both Query Shift (QS) and Query–Gallery Shift (QGS) settings across multiple datasets such as COCO-C, Flickr-C, Fashion-Gen, and Nocaps, which provides a reasonably comprehensive view of robustness across both synthetic corruption and cross-domain scenarios.

Weaknesses:
1. The GSR module is exclusively tailored for text. While text is discrete and highly sensitive to structural noise, image representations can also suffer catastrophic semantic shifts under severe corruptions (e.g., Pixelate, Zoom Blur). The paper achieves improvements in Image-to-Text retrieval (as seen in Table 1) solely through EPCL. It is a missed opportunity not to discuss or explore a visual analogue to semantic refinement (e.g., visual prompting or lightweight denoising).
2.  The authors somewhat downplay the latency introduced by the LLM. As reported in Table 4, the runtime increases by over 300% for T2I retrieval compared to the TCR baseline. In real-time retrieval systems, such latency is often unacceptable. The proposed "conditional fallback" is relegated to the conclusion without any empirical demonstration, weakening the argument that the method is practically deployable at scale.
3. Equation 9 constructs the negative set by checking $\mathcal{C}_i \cap \mathcal{E}_j \neq \emptyset$. This strict set intersection logic might fail if the base model's predictions are severely miscalibrated. If $q_j$ is highly noisy, its exclusion set $\mathcal{E}_j$ might mistakenly contain the true positives for $q_i$, turning valid targets into pushed-away negatives. The paper does not address how the method handles error propagation from severely corrupted queries affecting other queries in the batch.
4. Although the paper demonstrates improvements under multiple corruption settings, the analysis of failure cases is relatively limited. It would be helpful to understand scenarios where the GSR module produces incorrect semantic repairs or where EPCL introduces misleading exclusion constraints. Such analysis could provide deeper insight into the limitations of the framework.

---

> ### Author Rebuttal · Authors · 2026-03-31
>
> We thank the reviewer for the thorough assessment and valuable feedback.
>
> ---
>
> **W1 & Q4: Text-Only Refinement Limits Visual Robustness.**
>
> We agree a visual analogue to GSR could further improve I2T, but this is a deliberate trade-off. Unlike text, the image side does not only contain blur-like degradation, but 16 corruption types spanning (Noise, Blur, Weather, and Digital) categories. Therefore, a useful visual refinement module would need to handle a much broader corruption space than a single blur or denoising setting.
>
> In practice, there are two main options, both with clear limitations. A generative visual restoration model may better recover semantics across diverse corruptions, but it would be much slower. In contrast, a lightweight denoising/deblurring model is faster, but usually cannot cover such a wide range of corruption types well. In other words, we are not aware of a visual refinement module that is simultaneously fast enough and broad enough to handle all 16 image corruptions effectively.
>
> For this reason, we chose to keep the visual side lightweight and let EPCL handle image corruption in embedding space. Empirically, this design already improves Flickr-C I2T from 72.4\% to 74.8\% over TCR without additional runtime cost.
>
> ---
>
> **W2 & Q2: Efficiency Overhead from LLM-based GSR.**
>
> Due to space constraints, please refer to our response to **Reviewer J2wj (W1)** for a detailed discussion.
>
> ---
>
> **W3 & Q3: Sensitivity Analysis of Adaptation Steps and Stability under Noise.**
>
> We ablate the number of adaptation steps under the T2I Flickr-C setting with BLIP. Specifically, we fix the number of adaptation steps to $1, 3, 5$, respectively: step=1 yields R@1=65.2, while step=3 and step=5 both yield 65.7. Performance saturates at 3 steps, confirming that LN-only adaptation converges quickly and additional steps add computation without further gain. We therefore default to 3 steps for efficiency.
>
> For the second question, our current QS setting already evaluates ReEx under consecutive noisy queries, rather than isolated noisy samples. In Flickr-C T2I, following TCR, all incoming queries are corrupted, with the strongest severity levels applied to character-level / word-level / sentence-level perturbations. Therefore, the model is continuously adapted on a noisy stream throughout testing. Under this setting, ReEx still achieves $65.7\%$ average R@1 on Flickr-C, outperforming TCR by $2.3\%$, which suggests that the model does not exhibit obvious drift even when many noisy queries appear consecutively.
>
> Due to space constraints, please refer to our response to **Reviewer J2wj (W3)** for a detailed discussion on highly homogeneous test batches.
>
> ---
>
> **W4 & Q1: Robustness of GSR to Severe Corruption.**
>
> We additionally evaluate ReEx under varying noise intensities. Beyond the original QS settings (I2T: visual corruption level 5; T2I: text corruption level 7), we further test I2T under blur levels $1, 3, 5$ and T2I under character-level corruption levels $1, 3, 5, 7$.
>
> |Method|I2T-1|I2T-3|I2T-5|T2I-1|T2I-3|T2I-5|T2I-7|
> |---|---:|---:|---:|---:|---:|---:|---:|
> |TCR|89.5|84.1|72.4|75.1|62.7|48.5|31.3|
> |ReEx|**90.7**|**86.2**|**74.8**|**76.4**|**66.0**|**53.9**|**37.7**|
>
> The results show that both methods degrade as corruption becomes stronger, but ReEx degrades more slowly than TCR across all tested levels. In I2T, ReEx stays consistently above TCR by $1.2\%$ to $2.4\%$ from level 1 to 5. In T2I, the advantage becomes larger as corruption increases, growing from $1.3\%$ at level 1 to $6.4\%$ at level 7. This suggests that ReEx remains robust even when the query becomes highly ambiguous.
>
> This robustness is particularly meaningful at severity level 7, where corruption is already extreme. For example, under CI noise, *“A young female student performing a downward kick to break a board held by her Karate instructor.”* becomes *“A young female student perMfomriming a Fdo7wWnward dkVick to bhrbeak a @bo%ard hEelCd by her Karate inst6rqucytor.”* At this point, the sentence is already heavily damaged, so further increasing corruption is of limited practical value. Even so, ReEx improves T2I R@1 from 21.1% (TCR) to 33.4%, confirming GSR's robustness without severe semantic drift. Further gains under stronger corruption would require a more powerful LLM, leading to an accuracy–latency trade-off. More fundamentally, if corruption destroys semantic content beyond recovery, the bottleneck shifts from GSR to the TTA paradigm itself: any method is limited by the information remaining in the corrupted query, and methods lacking explicit semantic refinement face even tighter constraints.
>
> We will include representative visual failure examples with corresponding images in the appendix of the revised paper.

---

> > ### Author Rebuttal · Reviewer_XkQs · 2026-04-04
> >
> > My concerns have been adequately addressed by the authors. I will keep my positive score.

---

> > > ### Author Response · Authors · 2026-04-04
> > >
> > > Thank you for your positive feedback and for acknowledging that our responses have addressed your concerns. We sincerely appreciate your time and thoughtful evaluation of our work.

---

### Official Review · Reviewer_hFjf · 2026-03-08

**Soundness:** 3
**Presentation:** 3
**Significance:** 3
**Originality:** 3
**Overall Recommendation:** 5
**Confidence:** 5

**Summary:**

This paper proposes ReEx, a test-time adaptation framework for robust cross-modal retrieval under query shifts. The method incorporates Generative Semantic Refinement (GSR) to rectify corrupted queries using a lightweight LLM and fuse embeddings via Confidence-Guided Dynamic Fusion (CGDF), alongside Exclusion-guided Proxy Contrastive Learning (EPCL) which transforms low-confidence queries into effective supervisory signals by constructing candidate positive proxies and exclusion-based negative sets. Extensive experiments on COCO-C, Flickr-C under query shifts, and four query-gallery shift datasets show significant improvements over existing test-time adaptation methods.

**Compliance With Llm Reviewing Policy:**

Affirmed.

**Final Justification:**

The authors have address all my concern.

**Key Questions For Authors:**

1.	Given that the batch size directly dictates the scope of the candidate pool and exclusion sets, could the authors analyze the sensitivity of the ReEx for different batch sizes?
2.	Could the authors provide ablation results for EPCL in the I2T setting? This would help verify the effectiveness of exclusion-based constraints across modalities.
3.	The mathematical notation requires standardization. Unifying these symbols in the final version would improve clarity.

**Limitations:**

yes

**Strengths And Weaknesses:**

Strengths:
1. This paper shows that textual noise degrades retrieval performance and that existing test-time adaptation methods under query shift overemphasize high-confidence samples while overlooking signals inherent in low-confidence queries.
2. The proposed ReEx combines GSR with CGDF to mitigate semantic drift, and it shows that relying solely on LLM-based correction can introduce additional noise. Moreover, its exclusion-set-driven EPCL leverages the negative constraints in low-confidence queries to guide adaptation, offering an elegant and novel design.
3. The evaluation is comprehensive, showing state-of-the-art results on multiple benchmarks and reporting runtime overhead with performance–efficiency trade-offs across LLM sizes.

Weaknesses:
1. The current ablation studies are primarily conducted with a fixed batch size. Since batch size directly determines the candidate pool and exclusion-set sizes in EPCL, performance may be sensitive to this hyperparameter. The authors should report results across a range of batch sizes to better characterize this sensitivity.
2. Although the paper states that GSR is not employed in the I2T setting, this does not prevent a direct examination of EPCL’s contribution. Providing additional ablation studies for EPCL in the I2T setting would further demonstrate the effectiveness of exclusion-based negative constraints.
3. There are a few minor notation inconsistencies: embeddings are denoted in multiple ways (e.g., $z_{q,i}$ vs. $z_i$), and the candidate/exclusion set sizes alternate between $K$ and $k$.
4. Some cross-modal retrieval works are encouraged to be included, such as Visual Abstraction: A Plug-and-Play Approach for Text-Visual Retrieval, Long-clip: Unlocking the long-text capability of clip, Multi-granularity Correspondence Learning from Long-term Noisy Videos.

---

> ### Author Rebuttal · Authors · 2026-03-31
>
> We sincerely thank the reviewer for their thoughtful and constructive feedback. Below, we address each concern in detail:
>
> ---
>
> **W1 & Q1: Sensitivity Analysis of Batch Size.**
>
> Following the reviewer’s suggestion, we evaluate batch sizes $16/32/64$ on Flickr-C under the QS setting while keeping all other hyperparameters fixed. The results are shown below.
>
> | Batch size | I2T R@1 | T2I R@1 |
> |---|---:|---:|
> | 16 | **77.2** | 65.6 |
> | 32 | 76.2 | **65.7** |
> | 64 | 74.8 | **65.7** |
>
> The results show that the effect of batch size is different for the two retrieval directions. In T2I, the performance remains nearly unchanged across batch sizes, with R@1 varying only from $65.6\%$ to $65.7\%$, which indicates that ReEx is not sensitive to the scale of the candidate pool in this direction. In I2T, smaller batches lead to better performance, with R@1 decreasing from $77.2\%$ at batch size $16$ to $74.8\%$ at batch size $64$. A likely reason is I2T relies only on EPCL, while a larger batch brings a more diverse shared candidate pool and thus less accurate exclusion sets, which makes negative mining less effective. Overall, these results suggest that using a larger batch does not always improve ReEx. The best batch size depends on the retrieval direction: T2I stays stable across different batch sizes, whereas I2T works better with smaller batches.
>
> ---
>
> **W2 & Q2: Ablation Study of EPCL in the I2T Setting.**
>
> We additionally conduct an ablation study in the I2T setting to verify whether the exclusion-based constraint remains effective. Since GSR is not used in I2T, this analysis directly focuses on EPCL, and we therefore follow the ablation protocol in the main paper by separately examining its two components, CPA and ENM.
>
> | Method | I2T R@1 |
> |---|---:|
> | w/o ADAPT | 60.6 |
> | w/o CPA | 73.5 |
> | w/o ENM | 67.6 |
> | **ReEx** | **74.8** |
>
> The results show that the gain in I2T mainly comes from ENM. Compared with w/o ADAPT, ReEx improves R@1 from $60.6\%$ to $74.8\%$, confirming that test-time adaptation is essential under visual corruption. More importantly, removing ENM causes a large drop from $74.8\%$ to $67.6\%$, whereas removing CPA only reduces performance to $73.5\%$. This difference is consistent with the role of the two modules in our method. CPA builds a soft positive proxy from the candidate set, which mainly helps smooth local neighborhood noise. In contrast, ENM provides explicit negative constraints from both the exclusion set and contradictory queries in the current batch, directly refining the decision boundary. Therefore, these results directly verify that the exclusion-based constraint is effective not only in T2I, but also across modalities in I2T.
>
> ---
>
> **W3 & Q3: Standardization of Mathematical Notation.**
>
> We appreciate this suggestion and will make sure the final version is fully standardized for clarity. We will carefully revise the final version to standardize these definitions throughout the paper. Concretely, we will use a unified notation for embeddings, e.g., $z_{q,i}$, $\hat z_{q,i}$, and $\tilde z_{q,i}$ for the raw, rectified, and fused query embeddings, and $z_{g,j}$ for gallery embeddings, to avoid switching between forms such as $z_i$ and $z_{q,i}$. We will also use a single symbol, $K$, consistently for the sizes of both the candidate set and the exclusion set, which is already the intended definition in Eq. (7).
>
> ---
>
> **W4: Expansion of Related Work.**
>
> We thank the reviewer for the suggestions. We will cite these works and discuss them in the related work section of the revised manuscript.

---

> > ### Author Rebuttal · Reviewer_hFjf · 2026-04-01
> >
> > The authors have address all my concern.

---

> > > ### Author Response · Authors · 2026-04-01
> > >
> > > Thank you very much for raising your score and for your positive feedback. We sincerely appreciate your careful reading of our rebuttal and your recognition of our clarifications. Your constructive feedback has been invaluable in strengthening the paper.

---

### Official Review · Reviewer_J2wj · 2026-03-12

**Soundness:** 3
**Presentation:** 3
**Significance:** 2
**Originality:** 3
**Overall Recommendation:** 4
**Confidence:** 3

**Summary:**

This paper studies cross-modal retrieval with test-time adaptation in , with an emphasis on textual corruption and the underuse of low-confidence queries. It proposes Generative Semantic Refinement and Exclusion-Guided Adaptation, which combines Generative Semantic Refinement (GSR) to rectify structural textual noise via confidence-guided fusion, and Exclusion-Guided Proxy Contrastive Learning (EPCL) to mine informative negative constraints from ambiguous data, mitigating semantic drift and redefining decision boundaries.  Experiments are conducted on standard robustness benchmarks, such as COCO-C and Flickr-C.

**Compliance With Llm Reviewing Policy:**

Affirmed.

**Final Justification:**

I appreciate the author's rebuttal, and the current response largely addresses the raised concerns. I keep the initial recommendation unchanged (weak accept).

**Key Questions For Authors:**

1. The results demonstrate that ReEx achieves gains under query shift, particularly against text-side structural corruptions. However, under the query-gallery shift setting across multiple domain datasets, the performance gain over the baseline TCR is reduced. Could the authors clarify the specific shift where ReEx is expected to be most effective?
2. The empirical validation relies on a single pre-trained BLIP. Given that Vision-Language Pre-trained models have highly diverse architectural biases and feature spaces, how stable are the proposed modules across other architectures? (e.g., CLIP)
3. ReEx relies on a specific prompt template and predominantly evaluates lightweight LLMs (e.g., Qwen2.5-0.5B-Instruct). How sensitive is it to variations in prompt design, and how does the LLM handle domain-specific terminology where lightweight models are prone to hallucination?

**Limitations:**

yes

**Strengths And Weaknesses:**

Strengths:
-- It designs the Confidence-Guided Dynamic Fusion mechanism to repair structural noise while preventing LLM-induced semantic hallucinations.
-- The proposed dual modules Generative Semantic Refinement and Exclusion-Guided Proxy Contrastive Learning is highly self-consistent.
-- Extensive comparative and ablation experiments are conducted across large-scale benchmarks against character-, word-, and sentence-level corruptions.
-- This work is easy to follow and the techniques are clearly stated.
Weaknesses:
-- The proposed Generative Semantic Refinement module necessitates querying an LLM during inference, which drastically increases the latency from 0.24s to 0.75s. This severe 3x computational bottleneck contradicts the efficiency requirement of online TTA and the practical deployment in large-scale retrieval systems.
-- The empirical validation relies entirely on a single pre-trained BLIP. Given that Vision-Language Pre-trained (VLP) models have highly diverse architectural biases and feature spaces, the failure to evaluate the ReEx framework across other architectures (e.g., CLIP)
-- In the Exclusion-Guided Proxy Contrastive Learning (EPCL) module, the Exclusion Set  is constructed by selecting the "bottom-K" samples from a shared candidate pool derived from the current mini-batch. The manuscript lacks an analysis for scenarios where the incoming test batch is highly homogeneous (e.g., highly similar semantic concepts).
-- The source code isn’t provided, and the reproducibility is questionable.

---

> ### Author Rebuttal · Authors · 2026-03-31
>
> We thank the reviewer for the thorough assessment and valuable feedback.
>
> ---
>
> **W1: Efficiency Overhead from LLM-based GSR.**
>
> The extra latency comes entirely from GSR in T2I: with batch size 64, runtime increases from 0.24 s/batch to 0.75 s/batch, while I2T incurs no overhead. Our experiments apply GSR to all queries as an upper-bound evaluation under severe corruption, so this latency is real. The confidence-gated routing in the paper is a system-level optimization for average-case deployment: per-batch GSR latency remains fixed, but routing only low-confidence traffic to GSR reduces average system cost without applying GSR to all traffic. We will clarify this distinction in the revision.
>
> ---
>
> **W2 & Q2: Evaluation on Powerful VLP Backbone BLIP-2.**
>
> We additionally evaluate ReEx on BLIP-2-itm-vit-g. ReEx achieves **$67.1\%$** avg. R@1, **$1.5\%$** higher than TCR ($65.6\%$), confirming that our gains generalize to stronger VLP backbones. The full comparison table is provided in our response to **Reviewer 1Lew (W1 & Q3)**.
>
> ---
>
> **W3: EPCL under Homogeneous Test Batches.**
>
> In a highly homogeneous batch, the shared candidate pool is built from semantically similar queries, so cross-query negatives become weak or disappear. In this case, EPCL can degenerate to a simpler form: supervision shifts to the gallery-side candidate and exclusion sets, while the batch-wise exclusion term weakens. We view this as a graceful fallback rather than a failure mode, because forcing strong batch-wise separation in homogeneous batches risks introducing false negatives between semantically close queries. EPCL avoids this by preserving gallery-side exclusion and CPA-based proxy alignment as the primary objective. Improving homogeneous batch performance, e.g., via cross-batch memory or diversity-aware batching, is an interesting future direction.
>
> ---
>
> **W4: Reproducibility and Code Release.**
>
> We will release the code upon acceptance to ensure reproducibility.
>
> ---
>
> **Q1: Effectiveness under QS vs QGS.**
>
> In QS, query-side corruption directly matches our design: GSR repairs noisy semantics over a stable gallery, letting ReEx target the dominant error source. In QGS, the model is not fine-tuned on the target dataset and neither side has explicit corruption, shifting the bottleneck to cross-domain query–gallery misalignment. With no noisy semantics to repair, GSR contributes little. We provide new QGS T2I ablations to validate this:
>
> |Method|Flickr|COCO|Fashion|Nocaps ID|Nocaps ND|Nocaps OD|Avg.|
> |---|---|---|---|---|---|---|---|
> |w/o EPCL|67.1|45.0|24.7|74.6|63.4|67.3|57.0|
> |w/o GSR|70.2|**48.8**|29.2|**75.4**|65.3|68.8|59.6|
> |Full|**70.4**|48.6|**29.4**|75.2|**65.5**|**69.1**|**59.7**|
>
> Removing GSR drops avg. by only 0.1, confirming that GSR contributes little when the gallery has shifted. Removing EPCL drops avg. to 57.0, showing it remains the main contributor (+2.7%). However, EPCL's gain is still smaller than in QS, because under gallery-side shift, the candidate and exclusion sets become less reliable, making the negatives mined by EPCL less faithful to true cross-domain mismatches. The smaller QGS gain is not a failure of ReEx. It reflects that our design is most effective when the dominant shift lies on the query side, which is exactly the QS setting.
>
> ---
>
> **Q3: Sensitivity to Prompt Design and Domain-Specific Language.**
>
> We evaluate prompt sensitivity by replacing the original GSR prompt with three variants: (1) removing the semantic plausibility constraint, (2) using a minimal correction prompt, and (3) adding stronger restrictions to avoid unsupported rewriting.
>
> |Prompt variant|Avg. R@1|
> |---|---:|
> |Full prompt|**65.7**|
> |w/o semantic plausibility|65.3|
> |Minimal prompt|65.3|
> |Stronger restrictive prompt|65.0|
>
> ReEx is not sensitive to moderate prompt changes: removing the semantic plausibility constraint or using a minimal prompt only reduces Avg. R@1 from $65.7\%$ to $65.3\%$. The full prompt still performs best, suggesting semantic plausibility provides a small but consistent gain. The stronger restrictive prompt performs worse ($65.0\%$), because the current GSR uses a small LLM. Under severe corruption, overly strong instructions may be harder to follow while still recovering the correct meaning. Overall, the main gain comes from refinement + CGDF, rather than the exact prompt wording.
>
> Regarding domain-specific terminology, we agree this is a genuine limitation of lightweight LLMs and reflects an inherent efficiency–fidelity trade-off. Small LLMs are more likely to produce unsupported rewrites when encountering specialized expressions requiring stronger prior knowledge. ReEx mitigates this risk through CGDF, reducing over-correction when refinement is less reliable. We will discuss domain-specific refinement as an important future direction in the final version, such as distilling the corrector on domain-relevant noisy/clean pairs or using more conservative edit-based correction.

---

> > ### Author Rebuttal · Reviewer_J2wj · 2026-04-04
> >
> > I appreciate the additional clarifications and new experiments in the rebuttal, which resolve some of my questions regarding the QGS setting and prompt sensitivity, but they do not sufficiently change my overall assessment, so I keep my original score unchanged.
> > The paper would still benefit from a clearer discussion of the setting in which the method may be constrained or less effective, particularly regarding the unresolved computational bottleneck of the LLM-based GSR.

---

> > > ### Author Response · Authors · 2026-04-06
> > >
> > > Thank you for the helpful follow-up. We agree that the paper should discuss more clearly where ReEx is most effective and where it is constrained.
> > >
> > > ReEx is most beneficial when the dominant shift is on the query side, especially under text-side structural corruption, where GSR directly repairs the main error source and EPCL adapts on top of a relatively stable gallery. Its gain is naturally smaller when the main challenge becomes broader query-gallery misalignment, as reflected by the reduced improvement under QGS.
> > >
> > > We also agree that the LLM-based GSR is a real limitation of the current design. At the same time, this overhead reflects a trade-off: as shown in Table 4, the 0.75s/batch yields a +3.6% T2I gain over the baseline, which is nearly 3× TCR’s gain (+1.3%). In the revision, we will make this point more explicit and clarify that our current experiments apply GSR to all queries under severe corruption in order to highlight the robustness benefit under extreme query noise, rather than to represent a deployment-optimized setting.
> > >
> > > From a practical perspective, the more realistic usage of the current design is the confidence-gated routing. This is a system-level optimization for average-case deployment: the per-call latency of GSR remains unchanged once it is invoked, but routing only low-confidence queries to GSR can reduce the average system cost by avoiding LLM calls on the full query stream. We will clarify this distinction more explicitly in the revision.

---

### Official Review · Reviewer_1Lew · 2026-03-15

**Soundness:** 2
**Presentation:** 3
**Significance:** 2
**Originality:** 3
**Overall Recommendation:** 4
**Confidence:** 4

**Summary:**

This paper addresses the vulnerability of Vision-Language Pre-trained (VLP) models to real-world query noise in cross-modal retrieval tasks. The authors identify two critical limitations in existing Test-Time Adaptation (TTA) methods: (1) they rely on structurally corrupted text embeddings without explicit rectification, and (2) they discard low-confidence queries, overlooking valuable negative supervision signals. To address these issues, the paper proposes ReEx, a unified robust retrieval framework with two core modules:(1)Generative Semantic Refinement (GSR). (2)Exclusion-Guided Proxy Contrastive Learning (EPCL). The framework is evaluated on COCO-C and Flickr-C benchmarks under Query Shift and Query-Gallery Shift settings. ReEx achieves state-of-the-art results, outperforming the strong baseline TCR by 2.3% (T2I) and 2.4% (I2T) on Flickr-C. Ablation studies validate the critical synergy between GSR's structural rectification and EPCL's negative mining strategy.

**Compliance With Llm Reviewing Policy:**

Affirmed.

**Final Justification:**

Based on the comprehensive evidence provided in the rebuttal, I consider my concerns are fully resolved. I have decided to adjust my overall recommendation from "3: Weak reject" to "4: Weak accept". This adjustment is primarily due to the authors' new experiments on the powerful BLIP-2 backbone and the LLaMA-3-70B model, which effectively demonstrate the method's scalability and acceptable relative memory overhead, justifying an increase in both "Soundness" and "Significance" scores from "2: fair" to "3: good". Furthermore, the authors provided a reasonable explanation regarding the fundamental difficulty of formalizing theoretical bounds in non-stationary online TTA environments. I have kept the "Presentation" and "Originality" scores at "3: good" without unnecessary inflation, as they were already satisfactory.

**Key Questions For Authors:**

1. Provide formal theoretical analysis for ReEx’s core modules (e.g., EPCL loss convergence bounds, robustness guarantees for GSR’s confidence-guided dynamic fusion). If unavailable, elaborate on key challenges in formalizing these components.
2.Explain the stark performance gap (2.3–2.4% gain in Query Shift vs. 0.4% in QGS) from methodological and empirical perspectives (e.g., GSR/EPCL ineffectiveness under gallery distribution shift, EPCL exclusion set failure to capture cross-domain negative signals), and supplement with validating experiments.
3.Report experimental results of ReEx on SOTA VLP backbones (BLIP-2, CLIP) and large-scale LLMs (Llama3.2-7B). If performance gains hold/improve, confirm scalability; if degraded, clarify root causes and propose optimization strategies.
4.Present supplementary results of ReEx under gradient noise intensities (e.g., 10%/30%/50% character-level typos in text, mild/moderate/extreme visual blur). Clarify the noise intensity threshold where ReEx still outperforms baselines and where performance degrades.

**Limitations:**

No. (1) the over-reliance of GSR on LLMs (no lightweight/LLM-free alternative for resource-constrained edge devices); (2) the limited effectiveness of EPCL under joint query-gallery distribution shifts (the 0.4% modest gain in QGS setting); (3) the narrow evaluation scope (only BLIP backbone and small-scale LLMs, untested scalability on SOTA VLPs/larger LLMs); (4) the unaddressed robustness boundary of ReEx across gradient noise intensities. For each limitation, elaborate on its underlying technical causes and tentative future research directions to address it, rather than just a superficial mention.

**Strengths And Weaknesses:**

Strengths:
The work is technically rigorous with well-designed experiments covering Query Shift/Query-Gallery Shift settings on COCO-C/Flickr-C and cross-domain datasets. Exhaustive ablation studies, hyperparameter sensitivity analysis and computational complexity evaluation fully validate the effectiveness of each module, and the authors honestly acknowledge the LLM-induced latency limitation with practical mitigation strategies. All performance claims are backed by precise quantitative results and qualitative visualizations.

Weaknesses:
1.Lacks theoretical formalization (e.g., convergence proof of EPCL loss, robustness bounds of GSR fusion). Evaluates only BLIP as the VLP backbone and small-scale LLMs, with no scalability test on state-of-the-art VLPs/ larger LLMs. No performance evaluation across different noise intensities, leaving the robustness boundary unaddressed.
2.The innovation is incremental rather than paradigm-shifting, building on existing TTA/contrastive learning ideas without introducing new theoretical frameworks. Relies on a standard InfoNCE loss with minor modifications, lacking a novel loss function for low-confidence query learning. GSR is fully dependent on LLMs with no lightweight LLM-free refinement alternative for edge scenarios.

---

> ### Author Rebuttal · Authors · 2026-03-31
>
> We thank the reviewer for the thorough assessment and valuable feedback.
>
> ---
>
> **W1 & Q1: Theoretical Analysis of ReEx.**
>
> Our contribution is primarily algorithmic and empirical. Formal analysis is difficult because EPCL operates in an online TTA setting where candidate/exclusion sets are dynamically constructed from the evolving model state, making the objective coupled and non-stationary and violating standard i.i.d. assumptions. For GSR, the refinement involves a discrete external LLM while the final fusion depends on the gallery-dependent similarity landscape, making formal robustness bounds nontrivial. Nevertheless, both modules are explicitly designed for stability: EPCL uses confidence-aware soft proxies to remain conservative under ambiguity, and CGDF automatically reverts to the original query when LLM confidence is low. We note that the absence of formal guarantees is common in online TTA. None of the compared methods (TENT, SHOT, EATA, SAR, TCR) provide convergence proofs either, because the non-stationary setting fundamentally violates standard theoretical assumptions. Our extensive experiments across 31 corruption types empirically validate this stability. We will add the above analysis to the final version.
>
> ---
>
> **W1 & Q3 & L3: Evaluation on Powerful VLP Backbone BLIP-2.**
>
> We additionally evaluate ReEx on BLIP-2-itm-vit-g. ReEx achieves $67.1\%$ avg. R@1, $1.5\%$ higher than TCR, confirming that our gains transfer to stronger backbones.
>
> |Method|I2T|T2I|Avg.|
> |---|---:|---:|---:|
> |baseline|65.0|56.9|61.0|
> |Tent|64.8|56.8|60.8|
> |SHOT|72.4|58.3|65.4|
> |EATA|65.0|57.4|61.2|
> |SAR|64.9|56.9|60.9|
> |READ|64.9|57.7|61.3|
> |DeYO|64.9|57.4|61.2|
> |TCR|72.0|59.1|65.6|
> |**ReEx**|**73.1**|**61.0**|**67.1**|
>
> ---
>
> **W1 & Q4 & L4: Robustness under Different Noise Intensities.**
>
> Note that our original QS setting already uses the strongest corruption levels (visual level 5 for I2T, character-level 7 for T2I). We further test across different corruption severities:
>
> |Method|I2T-1|I2T-3|I2T-5|T2I-1|T2I-3|T2I-5|T2I-7|
> |---|---:|---:|---:|---:|---:|---:|---:|
> | TCR |89.5|84.1|72.4|75.1|62.7|48.5|31.3|
> |ReEx|**90.7**|**86.2**|**74.8**|**76.4**|**66.0**|**53.9**|**37.7**|
>
> ReEx consistently outperforms TCR, and the advantage grows with corruption severity (T2I: +1.3% at level 1 → +6.4% at level 7). This suggests that ReEx remains robust even when the query becomes highly ambiguous.
>
> ---
>
> **W2: Limited Novelty Beyond Existing Contrastive TTA.**
>
> Our contribution is not a new contrastive  loss, but a new learning mechanism for low-confidence queries in online cross-modal TTA. Prior methods mainly rely on high-confidence positives, while ReEx turns ambiguous queries into usable supervision through Candidate/Exclusion set construction, soft proxy alignment (CPA), and exclusion-aware negative mining (ENM). Thus, the key novelty lies in how supervision is defined for low-confidence queries, not in replacing InfoNCE with another formula. Moreover, ReEx is not only EPCL. It also introduces GSR + CGDF to explicitly repair structurally corrupted text before adaptation, which is distinct from existing TTA methods that adapt directly on noisy embeddings.
>
> ---
>
> **W2 & L1: Dependence on LLM-based GSR without Lightweight Alternatives.**
>
> We agree this is a practical limitation. GSR solves semantic reconstruction, not simple spell-checking: under character-level/OCR-like noise, it must recover missing words, repair structure, and preserve semantics simultaneously, which is why rule-based correctors are insufficient. We note that the current LLM overhead is a deliberate accuracy–latency trade-off: as shown in Table 4, the 0.75s/batch cost yields a +2.3% T2I gain over TCR, nearly tripling TCR's own gain over the baseline (+1.3%).
>
> In practical deployment, two concrete directions can further reduce this cost. First, confidence-gated routing: bypass GSR for high-confidence queries and invoke it only when confidence falls below a threshold, reducing average latency since most queries do not require rectification. Second, edit-based distillation: use the LLM to generate (noisy, clean) pairs offline, then distill a lightweight edit predictor for sparse token edits rather than full rewrites, amortizing LLM cost into a one-time step. Both are engineering-feasible and compatible with our framework without modifying the core algorithm.
>
> ---
>
> **Q2 & L2: Limited Effectiveness under Query-Gallery Shift.**
>
> In QS, the gallery is stable and corruption lies on the query side, directly matching GSR+EPCL's design, so ReEx can address the dominant error source and achieve larger gains. In QGS, the mismatch exists on both sides, so the bottleneck shifts to query–gallery feature misalignment rather than query corruption, making query-side refinement less effective. We provide new QGS T2I ablations (see table in our response to **Reviewer J2wj (Q1)**, omitted here due to space constraints).

---

> > ### Author Rebuttal · Reviewer_1Lew · 2026-04-04
> >
> > All concerns are well addressed.

---

> > > ### Author Response · Authors · 2026-04-05
> > >
> > > We sincerely thank the reviewer for raising the score and for acknowledging that our rebuttal has addressed all concerns. We greatly appreciate the reviewer’s time and careful reconsideration of our work.

---

### Decision · Program_Chairs · 2026-04-30

**Decision:**

Accept (regular)

**Comment:**

This paper addresses an important problem in robust cross-modal retrieval under test-time query corruption and distribution shift. It proposes ReEx, a unified framework that combines GSR for repairing noisy textual queries and EPCL for exploiting low-confidence queries as informative supervision. The paper’s main strengths are its clear motivation, coherent technical design, and strong empirical validation. The rebuttal further improved the submission by adding experiments on stronger backbones and larger language models, thereby increasing confidence in the scalability and generality of the approach while resolving several key concerns. The main remaining weakness is efficiency, as the LLM-based refinement introduces additional inference latency. Overall, I believe the strengths outweigh the weaknesses, and I therefore recommend Accept.